# A non-genotoxic stem cell therapy boosts lymphopoiesis and averts age-related blood diseases in mice

Anna Konturek-Ciesla ●[1,2], Qinyu Zhang ●[1], Shabnam Kharazi[1] & David Bryder ●[1] ✉

Hematopoietic stem cell (HSC) transplantation offers a cure for a variety of blood disorders, predominantly affecting the elderly; however, its application, especially in this demographic, is limited by treatment toxicity. In response, we employ a murine transplantation model based on low-intensity conditioning protocols using antibody-mediated HSC depletion. While aging presents a significant barrier to effective HSC engraftment, optimizing HSC doses and non-genotoxic targeting methods greatly enhance the long-term multilineage activity of the transplanted cells. We demonstrate that young HSCs, once effectively engrafted in aged hosts, improve hematopoietic output and ameliorate age-compromised lymphopoiesis. This culminated in a strategy that robustly mitigates disease progression in a genetic model of myelodysplastic syndrome. These results suggest that non-genotoxic HSC transplantation could fundamentally change the clinical management of age-associated hematological disorders, offering a prophylactic tool to delay or even prevent their onset in elderly patients.

Bone marrow transplantation (BMT) is a curative treatment for numerous blood and immune diseases[1]. BMT works by introducing healthy donor hematopoietic stem cells (HSCs) to individuals with defective or damaged hematopoiesis. These HSCs, in turn, regenerate the entire hematopoietic system and assure life-long hematopoiesis via their dual capacity for multilineage differentiation and self-renewal[2].

Despite the broad therapeutic potential of BMT, its adoption is constrained by health complications from current transplantation procedures. Successful BMT requires conditioning, which typically involves administering cytotoxic chemotherapy and/or total body irradiation (TBI). Conditioning eliminates host cells and provides space for newly transplanted HSCs, but comes at the price of deleterious side effects[3]. These include changes to the bone marrow (BM) architecture that may lead to long-term residual hematopoietic injury[4] and influence the fate of the transplanted HSCs[5].

To address the shortcomings of traditional conditioning regimens, alternative strategies have been developed to selectively eliminate hematopoietic stem and progenitor cells (HSPCs) from the BM while preserving non-hematopoietic cells. These strategies include the use of monoclonal antibodies to block essential survival signals for HSPCs[6] or to deliver lethal payloads specifically to these cells[7]. Additional non-genotoxic approaches, such as mobilization-based regimens, use agents like granulocyte colony-stimulating factor (G-CSF) and AMD3100 to disrupt HSPC-niche interactions, thereby vacating niches for transplanted cells[8]. Another method involves overcoming transplantation barriers by using higher doses of HSPCs[9–12].

Aging has profound effects on hematopoiesis that lead to an increased predisposition to a range of hematological shortcomings, including myelodysplasia and anemia[13]. Aging is associated with a decreased proportion of naive *B* and *T* cells and a corresponding increase in the frequency of memory-type *B* and *T* cells, which likely contribute to diminished immune responses to new antigens[14]. The decline in de novo production of new lymphocytes with age can, at

[1]Division of Molecular Hematology, Department of Laboratory Medicine, Lund Stem Cell Center, Medical Faculty, Lund University, Lund, Sweden. [2]Department of Biosystems Science and Engineering, ETH Zurich, Basel, Switzerland. ✉e-mail: David.Bryder@med.lu.se

least partially, be attributed to intrinsic changes accompanying the aging of HSCs[15,16]. Strategies aimed at re-instating more youthful hematopoiesis in aged individuals have, therefore, aimed to alter the function of HSCs in the aged setting[17–19]. However, such approaches often lead to only partial rejuvenation of aged HSCs. On the other hand, transplantation of young HSCs into aged hosts offers an opportunity to re-establish the entire hematopoietic system with young-like features. Practical limitations for this include an inability of aged individuals to cope with the devastating side effects of cytotoxic conditioning and the unresolved impact of host age/environment on graft fate. Previous studies have, for instance, proposed that recipient age decreases the efficiency of homing and long-term engraftment of transplanted HSCs[20,21], perhaps because of a more hostile pro-inflammatory BM microenvironment with age[22].

Here, we employed non-genotoxic transplantation methods to assess how recipients' age affects transplantation success. While several key challenges needed to be overcome, we demonstrate the successful reinstatement of multilineage hematopoiesis from young HSCs in aged recipients. Finally, we present the potential of non-genotoxic conditioning to prevent the emergence of hematological malignancy in a model for myelodysplastic syndrome.

## Results

### The aged BM environment restrains HSC engraftment

CD45-saporin (CD45-SAP) has previously been shown to be an effective non-genotoxic conditioning regimen in murine models of genetic disorders, such as immunodeficiency[23] and sickle cell anemia[7] in young animals. However, its efficacy has not yet been tested in older subjects. Therefore, our initial investigation aimed to evaluate the efficacy of CD45-SAP conditioning[7] in the context of aged hosts. Following administration of CD45-SAP (3 mg/kg) to young (2 months) and aged (16 months) C57BL/6-CD45.2 mice and analysis 8 days later (Fig. 1a), we observed only marginal changes in overall peripheral blood (PB) white blood cell (WBC) counts in both groups (Fig. 1b). More detailed assessments revealed transient reductions in platelets and hemoglobin of CD45-SAP-treated mice (Supplementary Fig. 1a) and some noticeable changes in WBC distribution, with reduced lymphocyte and elevated myeloid cell counts (Supplementary Fig. 1b).

While splenic cellularity remained relatively constant following CD45-SAP-treatment, more evident reductions were observed in thymic cellularity (Fig. 1c). The overall BM cellularity was also relatively unchanged (Fig. 1d, *left*), while in agreement with other studies[7,24], we observed a pronounced decrease in the numbers of HSCs (Fig. 1d, *right*). In young mice, the effects on other multipotent and more lineage-restricted BM progenitors varied in a cell-type-specific manner,

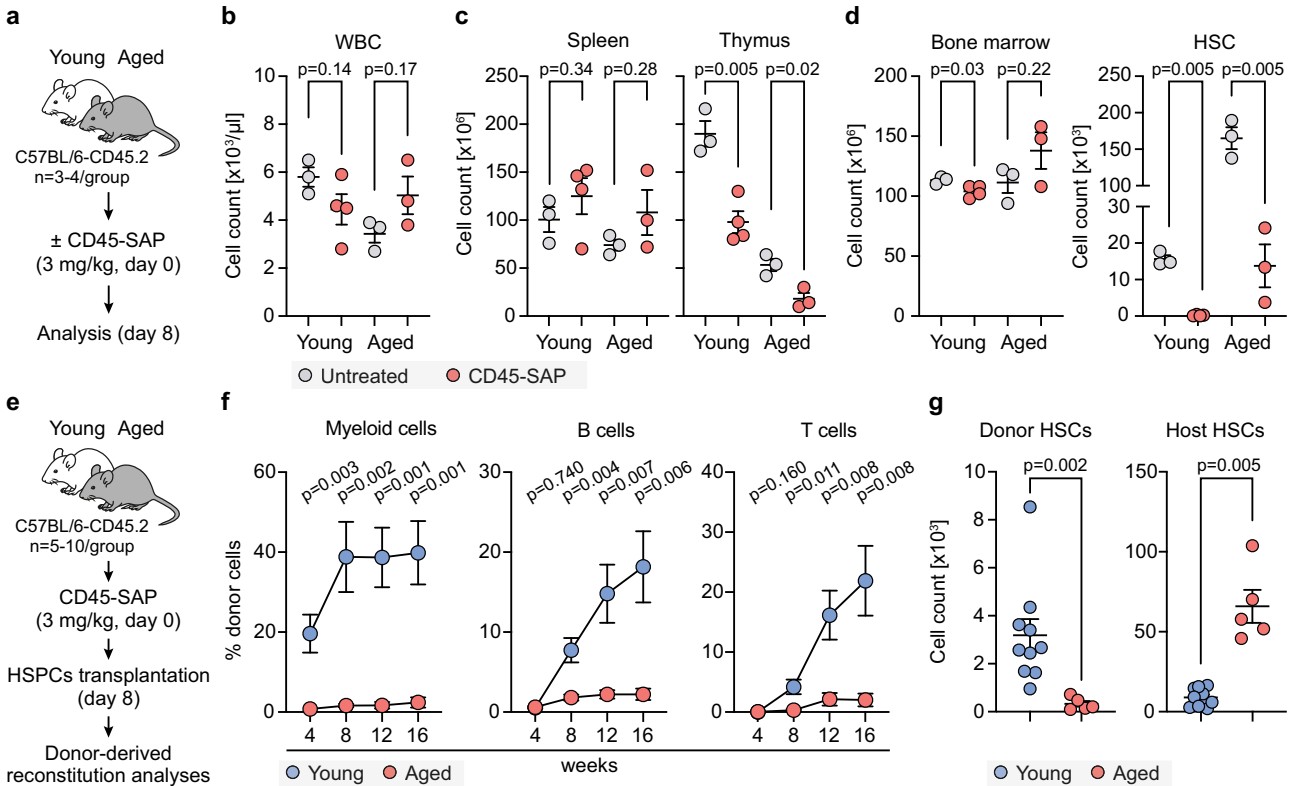

**Fig. 1 | The aged bone marrow environment restrains HSC engraftment.**
**a** Experimental design for Fig. 1b–d. Young (2 months, *n* = 4) and aged (16 months, *n* = 3) C57BL/6-CD45.2 mice received an intravenous injection of CD45-SAP (3 mg/kg), and hematopoietic cell subsets were analyzed after 8 days. Untreated mice served as controls (*n* = 3 for both young and aged groups). **b** Peripheral blood (PB) white blood cell (WBC) counts in untreated and CD45-SAP-treated mice. **c** Spleen and thymic cellularity in untreated and CD45-SAP-treated mice. **d** Bone marrow cellularity (*left*) and absolute HSCs numbers (*right*) in young and aged untreated and CD45-SAP-treated mice. In (b–d), gray and red circles represent untreated and CD45-SAP-treated groups, respectively. **e** Experimental design for Fig. 1f–g. Young (2 months, *n* = 10) and aged (16 months, *n* = 5) C57BL/6-CD45.2. mice received

CD45-SAP (3 mg/kg). Eight days after treatment, the mice were transplanted with HSPCs derived from young (2–4 months) mice. **f** Donor-derived reconstitution in PB of young and aged recipients. Two-group comparison at individual time-points was done using an unpaired two-sided *t* test with Welch correction. **g** Absolute numbers of donor- (*left*) and host-derived (*right*) HSCs in young and aged mice 17 weeks after transplantation. In (f–g), blue and red circles represent young and aged groups, respectively. In (b-d) and (g), points represent individual mice. Shown is the mean ± SEM. Statistical significance was determined using an unpaired two-sided *t* test with Welch correction. See also Supplementary Table 1. Source data are provided as a Source Data file.

but many of these changes were less pronounced in aged mice (Supplementary Fig. 1c).

We next transplanted CD45-SAP-treated young and aged mice with HSPCs derived from young mice (Fig. 1e). Donor-derived reconstitution was monitored in PB and by analyzing HSC chimerism in the BM at the experimental endpoint. While young recipients were effectively reconstituted, aged mice presented with only marginal donor-derived reconstitution in the PB (Fig. 1f and Supplementary Fig. 1d). This was further reflected in the levels of donor-derived HSCs (Fig. 1g).

To enhance reconstitution levels in aged hosts, we explored additional conditions in which CD45-SAP was co-injected with other selective immunotoxins and antibodies. These included treating aged mice with CD45-SAP in combination with CD8-SAP, CD4-SAP, or a *B* cell-specific antibody cocktail (Supplementary Fig. 1e). However, neither of these treatments led to any evident enhancement in donor cell engraftment (Supplementary Fig. 1f).

We also expanded our investigation to include a combinatorial treatment with CD45-SAP and low-dose (200 cGy) TBI in young and aged mice (Supplementary Fig. 1g). This aimed to understand whether this synergistic approach could enhance HSC engraftment while minimizing the toxic effects of higher-dose TBI. While this effectively enhanced the reconstitution levels in young recipients to achieve near-complete donor-derived chimerism, this approach was much less effective for aged mice (Supplementary Fig. 1h).

In summary, these results show that immunotoxin-based conditioning is less effective in advanced age, hindering HSC engraftment and transplantation success in C57BL/6 mice.

## Ex vivo expanded HSCs effectively reconstitute multilineage hematopoiesis in young CD45-SAP-conditioned recipients

Previous work by Wilkinson et al. established the efficacy of a polyvinyl alcohol (PVA)-based culture system in promoting murine HSC expansion[11], which we have validated in our prior work[12]. Notably, expanded HSCs enable a degree of HSC-derived reconstitution even in completely unconditioned hosts[11,12]. Given this, we explored the impact of larger quantities of HSCs on in vivo reconstitution outcomes in both non-conditioned and alternatively conditioned hosts.

We expanded HSCs ex vivo for 21 days and transplanted equivalent fractions (EE) derived from either 100 or 500 HSCs into unconditioned young hosts (Fig. 2a). Existing literature suggests that the BM niches available for engraftment in unconditioned hosts are limited, yet they are continuously made accessible through a process of niche recycling[9]. With this concept in mind, we examined the reconstitution outcome when the EE500 was subdivided into five separate fractions. Each of these fractions was then transplanted at weekly intervals to evaluate the possible advantage of spreading the transplantation over time (Fig. 2a).

In agreement with previous work[11], we observed that all non-conditioned recipients of ex vivo expanded HSCs demonstrated durable long-term multilineage engraftment, although the lymphoid chimerism, and in particular for the B cell lineage, was not on par with the myeloid reconstitution (Fig. 2b). EE500 resulted in higher engraftment compared to EE100, demonstrating a linear increase in myeloid lineage chimerism ($19.0 \pm 2.1$ vs. $4.3 \pm 2.0$) (Fig. 2b). However, dividing the EE500 graft into five weekly doses did not yield better results than a single bolus injection (Fig. 2b).

Subsequently, we evaluated the performance of ex vivo expanded HSCs in young hosts conditioned with CD45-SAP, comparing their behavior with that of hosts subjected to lethal (950 cGy) TBI (Fig. 2c). As anticipated, lethal TBI led to near-complete multilineage donor-reconstitution (Fig. 2d). CD45-SAP conditioning also resulted in prominent multilineage reconstitution, albeit with a lesser contribution to lymphoid lineages (Fig. 2d). This was further reflected in the PB lineage distribution of donor-derived cells (Supplementary Fig. 2a). Crucially, an examination of HSC chimerism at the end of the experiment demonstrated reconstitution levels equivalent to those observed for myeloid lineages (Fig. 2e), reinforcing that myeloid reconstitution serves as a dependable measure of ongoing HSC activity[25].

HSC transplantation into TBI-conditioned hosts detrimentally impacts their capacity to reconstitute secondary hosts[26]. To ascertain whether this also holds true for CD45-SAP-conditioned hosts, we conducted secondary transplantations of BM cells from the primary transplanted CD45-SAP-conditioned hosts. Two scenarios were considered: a) a non-competitive context where the transplanted cells in secondary hosts competed with the endogenous HSCs from the primary hosts, and b) a situation where the transplanted cells were mixed with an equal number of BM cells from young, untreated mice. BM cells harvested from primary TBI-treated recipients were included for comparison (Fig. 2f). This disclosed that high reconstitution levels observed in primary CD45-SAP-treated hosts were sustained in the non-competitive setting (Fig. 2g). Conversely, the reconstitution levels were markedly decreased upon competitive transplantation, mirroring the reduction in HSC activity derived from primary TBI-conditioned hosts (Fig. 2h, Supplementary Fig. 2b).

In summary, these results affirm prior studies, underscoring that while unconditioned wild-type (WT) hosts can attain long-term HSC-derived multilineage reconstitution, this necessitates significant quantities of HSCs[10]. Notably, pairing higher doses of HSCs with CD45-SAP conditioning dramatically enhanced the reconstitution outcomes. However, an intriguing parallel was noted with HSCs transplanted into TBI-conditioned hosts, where the process of transplantation itself appears to impair their potential for serial transplantation.

## Engraftment efficiency and functionality of transplanted young HSCs are maintained in aged hosts

The interplay between HSCs and their BM niche encompasses intricate physiological interactions[27]. This complexity might be amplified in the transplantation setting, considering the dynamics between donor HSCs and the host environment. Given the reduced engraftment of young HSCs in aged mice (Fig. 1), we explored whether an adverse environment in aged recipients, alongside ineffective conditioning, may contribute to this barrier.

To approach these questions experimentally, we harvested HSCs from young mice, expanded them ex vivo, and labeled the expanded grafts with Cell Trace Violet (CTV) dye. The CTV-labeled cells were then transplanted into both unconditioned and CD45-SAP-conditioned young and aged hosts. This allowed for the assessment of engraftment and proliferation of CTV-labeled young HSCs 2–4 weeks post-transplantation (Fig. 3a).

Analyses of unconditioned hosts revealed that young-derived HSCs could be recovered from both young and aged recipients, but with a tendency for lower efficiency in aged hosts (1.5-fold, Fig. 3b). More striking, but in agreement with the well-established expansion of HSCs associated with murine aging[2], aged recipients exhibited a significantly increased number of host HSCs in the unconditioned setting (Fig. 3b).

When analyzing CD45-SAP-conditioned young and aged hosts, we did not observe marked differences in the amounts of recoverable young donor HSCs (Fig. 3b). This suggests that compromised homing/engraftment in aged mice was unlikely to explain the inefficient reconstitution from young HSCs (Fig. 1). While CD45-SAP conditioning reduced the numbers of host HSCs in both young and aged mice (5.6- and 8.6-fold, respectively), aged recipients still harbored a notably higher number of host HSCs than young recipients (Fig. 3b).

The CTV dye dilution analysis in an unconditioned setting revealed that $21.1 \pm 3.3\%$ of donor HSCs remained undivided in young hosts, as compared to $30.9 \pm 5.4\%$ in aged recipients. In contrast, CD45-SAP treatment notably accelerated donor HSC proliferation, with similar effects observed across both age groups (Fig. 3c, d).

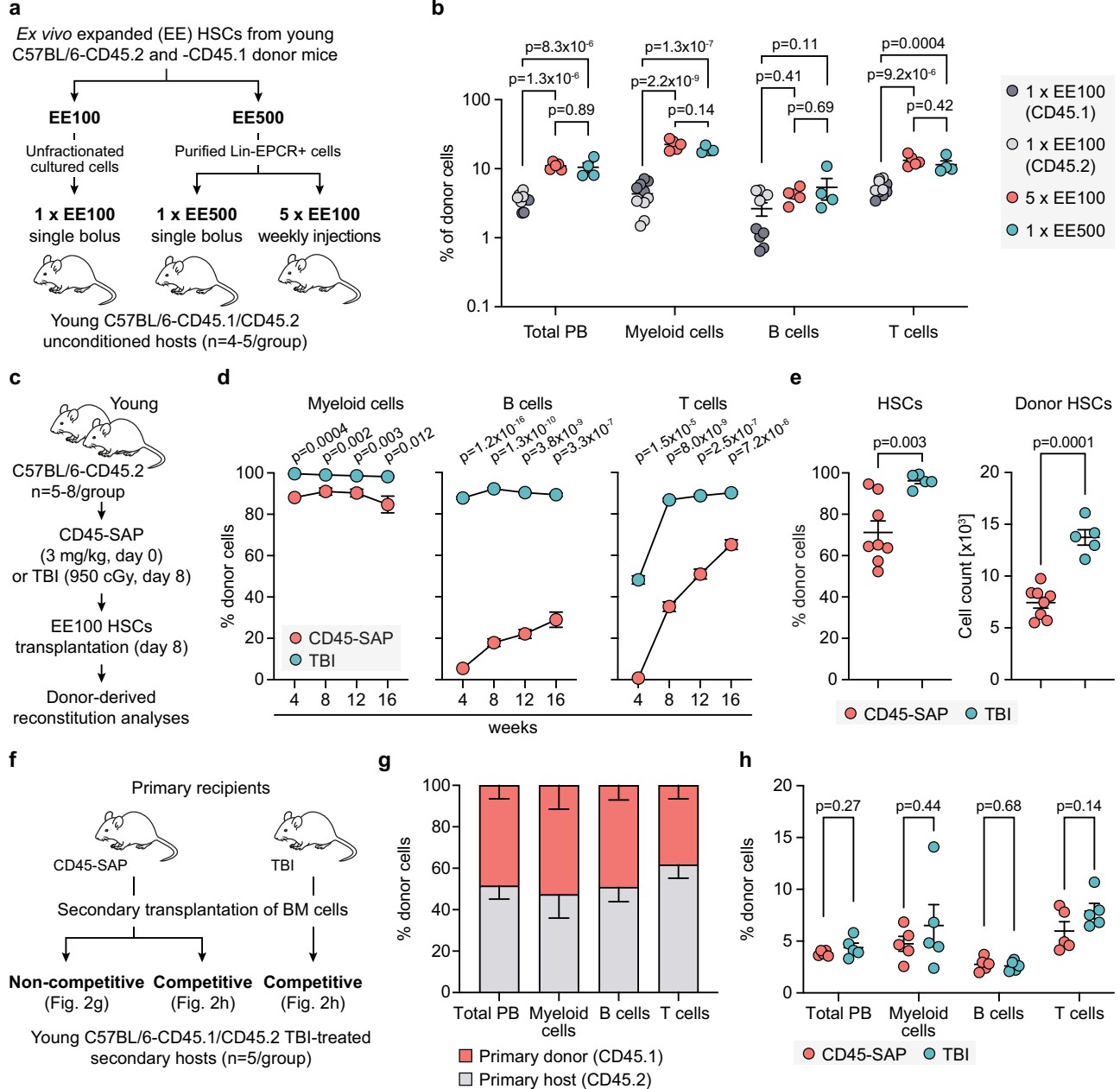

**Fig. 2 | Ex vivo expanded HSCs effectively reconstitute multilineage hematopoiesis in young CD45-SAP-conditioned recipients. a** Experimental design for Fig. 2b. HSCs were isolated from young (2–3 months) C57BL/6-CD45.2 and C57BL/6-CD45.1 mice, ex vivo expanded for 21 days and transplanted into young (3–5 months) unconditioned C57BL/6-CD45.1/CD45.2 recipients. Total cultures with equivalent expansions from 100 input HSCs were transplanted via a single injection (1 × EE100, n = 10). For 5 × EE100 and 1 × EE500, fractions of Lineage-EPCR+ cells from cultures initiated with 100 or 500 input HSCs were transplanted over five weekly injections (n = 5) or via a single injection (n = 4), respectively. **b** Donor-derived reconstitution from EE100 and EE500 cultured HSCs in indicated PB lineages 16 weeks after transplantation (1 × EE100, n = 10, gray circle; 5 × EE100, n = 5, red circle; 1 × EE500, n = 4, turquoise circle). Statistical significance was determined by one-way ANOVA with a Tukey post-hoc test. **c** Experimental design for Fig. 2d–e. 100 HSCs were isolated from C57BL/6-CD45.1 mice, expanded ex vivo for 21 days, and transplanted into young (2–3 months) C57BL/6-CD45.2 recipients treated with CD45-SAP (3 mg/kg, n = 8) or total body irradiation (TBI, 950 cGy, n = 5). For TBI-treated mice, cultured cells were transplanted together with 500,000 BM cells from C57BL/6-CD45.2 mice. **d** Reconstitution kinetics for

indicated PB lineages in CD45-SAP and TBI-treated mice. **e** Frequency (*left*) and absolute numbers (*right*) of donor-derived HSCs in CD45-SAP and TBI-treated mice 20 weeks after transplantation. **f** Experimental design for Fig. 2g–h. BM cells from primary recipients (Fig. 2e) were serially transplanted into young (2–3 months) TBI-treated C57BL/6-CD45.1/CD45.2 secondary hosts. For non-competitive transplantation, 3 × 10[6] BM cells from CD45-SAP-treated primary recipients were transplanted (n = 5). In a competitive setting, donor cells were mixed with competitor BM-derived from C57BL/6-CD45.1/CD45.2 mice at a 1:1 ratio prior to transplantation (n = 5 for both CD45-SAP and TBI-treated primary hosts). **g** Donor-derived PB reconstitution 16 weeks after non-competitive secondary transplantation of BM cells from primary CD45-SAP-treated hosts. **h** Donor-derived PB reconstitution 16 weeks after competitive secondary transplantation of BM cells from CD45-SAP or TBI-treated primary hosts. In (**b**), (**e**), and (**h**), points represent individual mice. In (d), (e), and (h), an unpaired two-sided *t* test with Welch correction was used. Red and turquoise circles represent CD45-SAP- and TBI-treated groups, respectively. In (**b, d, e, g,** and **h**), shown is the mean ± SEM. See also Supplementary Table 1. Source data are provided as a Source Data file.

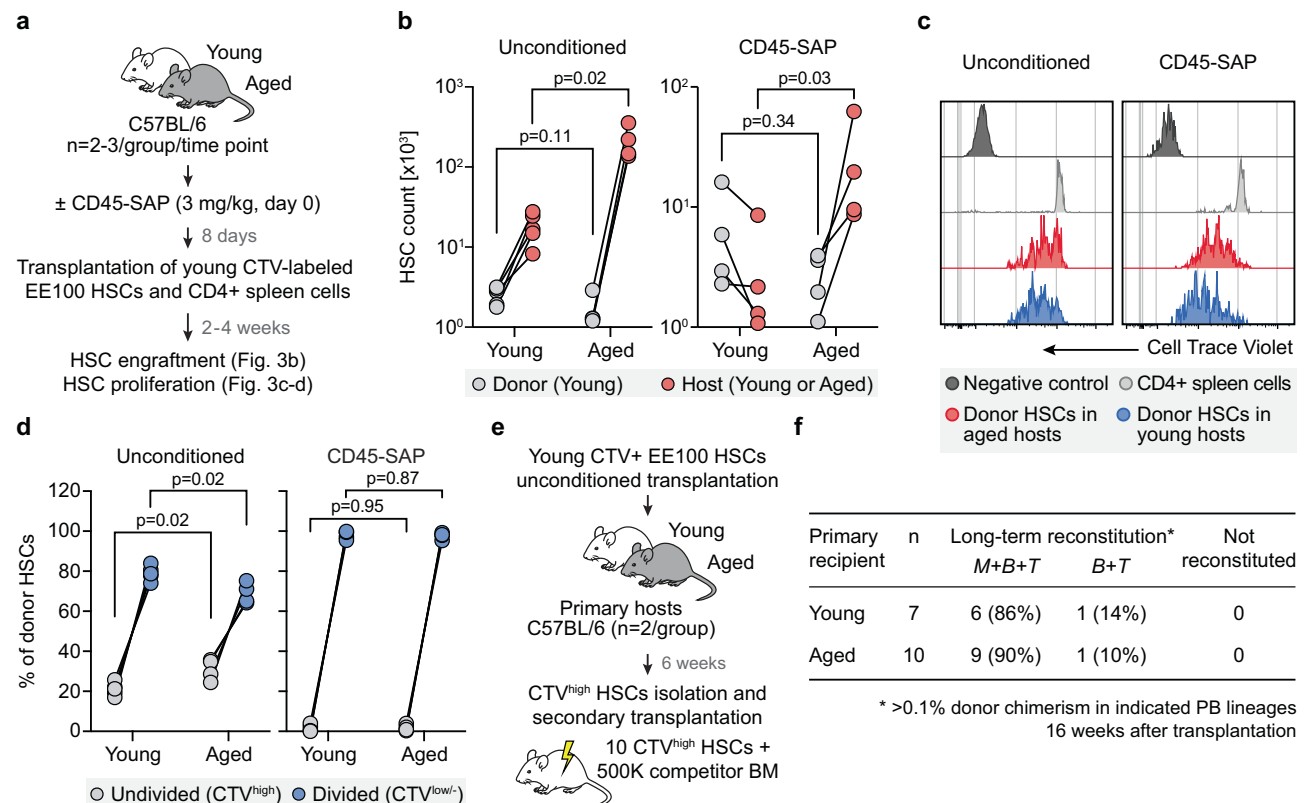

**Fig. 3 | The engraftment efficiency and functionality of transplanted young HSCs are maintained in aged hosts. a** Experimental design for Fig. 3b–d. HSCs were isolated from young (2–3 months) mice and ex vivo expanded for 21 days. Cultured cells and CD4+ spleen cells were next labeled with Cell Trace Violet (CTV) dye and co-transplanted into unconditioned or CD45-SAP-treated young (2–3 months, n = 5) and aged (16–17 months, n = 4) recipients at EE100 cells/mouse. HSC engraftment and CTV labeling were analyzed 2–4 weeks after transplantation. The groups were analyzed at different time points, with the CTV-positive signal in HSCs being determined based on the reference CD4+ spleen cells. **b** Absolute numbers of donor and host HSCs in unconditioned (*left*) and CD45-SAP-treated (right) young and aged recipients. Statistical significance was determined using the two-sided Mann-Whitney $U$ test. Gray and red circles represent donor (young) and host (young or aged) groups, respectively. **c** Representative histograms depicting CTV labeling in donor-derived HSCs in unconditioned (*left*) and CD45-SAP-treated (right) young and aged recipients 2 weeks after transplantation. **d** Frequency of undivided and divided donor HSCs in unconditioned (*left*) and CD45-SAP-treated (right) young and aged recipients. Statistical significance was determined using an unpaired two-sided $t$ test with Welch correction. Gray and blue circles represent undivided (CTV$^{high}$) and divided (CTV$^{low/-}$) groups, respectively. **e** Experimental design for Fig. 3f. HSCs were isolated from young (2–3 months) mice and ex vivo expanded as in (**a**). Cultured cells were next labeled with CTV dye and transplanted into unconditioned young (2–3 months, n = 2) and aged (16–17 months, n = 2) recipients at EE100 cells/mouse. Undivided HSCs were extracted from primary hosts 6 weeks after transplantation and competitively transplanted into young (2 months) TBI-treated secondary recipients (n = 7 for young and n = 10 for aged primary hosts). **f** Hematopoietic reconstitution from HSCs isolated from young and aged primary recipients. Values in parenthesis indicate the frequency of mice with multilineage (M + B + T) and lymphoid (B + T) reconstitution 16 weeks after transplantation. In (**b**) and (**d**), points indicate values for individual mice. See also Supplementary Table 1. Source data are provided as a Source Data file.

To further evaluate the functionality of young HSCs exposed to an aged environment, we isolated HSCs from young mice, expanded them ex vivo, and labeled them with CTV dye. We then transplanted these cells into unconditioned young and aged mice. Six weeks later, CTV-positive HSCs were extracted from the primary hosts and competitively transplanted into TBI-conditioned recipients (Fig. 3e). These experiments demonstrated efficient long-term multilineage reconstitution from isolated HSCs, irrespective of whether the cells were obtained from young or aged primary hosts (Fig. 3f). Interestingly, the overall chimerism was notably higher in recipients of cells derived from aged primary hosts (Supplementary Fig. 2c).

Together, these results demonstrate that young HSCs can successfully engraft in an aged environment with a preserved capacity for long-term multilineage reconstitution.

## Young HSCs support youthful hematopoietic characteristics upon transplantation into aged recipients

Although young HSCs successfully reconstituted aged recipients, the reconstitution was limited (Fig. 3). We hypothesized that residual endogenous HSCs, resulting from ineffective CD45-SAP conditioning, might restrict this process (Fig. 3b). Therefore, more thorough elimination of the host's aged HSCs could potentially enhance hematopoiesis from transplanted young HSCs.

Recent studies suggest that mobilizing endogenous HSCs could be a viable strategy to coax these cells out of their supportive niches within the BM, thereby creating vacant niches for transplanted HSCs[8,28]. Therefore, our subsequent experiments evaluated the reconstitution levels of EE100 young HSCs after CD45-SAP conditioning, either alone or in combination with a G-CSF/AMD3100-based mobilization protocol (Fig. 4a).

Examination of PB parameters 18 weeks post-transplantation revealed comparable WBC counts across untreated and differentially conditioned mice (Supplementary Fig. 3a). In contrast, platelet and hemoglobin levels differed between the groups, with CD45-SAP/G-CSF/AMD3100-treated recipients exhibiting markedly elevated hemoglobin levels (Supplementary Fig. 3b-c). Detailed evaluation of donor-derived reconstitution showed that aged mice undergoing the combined CD45-SAP/mobilization-based conditioning had over a two-fold increase in donor-derived multilineage reconstitution compared to those conditioned with CD45-SAP alone (Fig. 4b). This

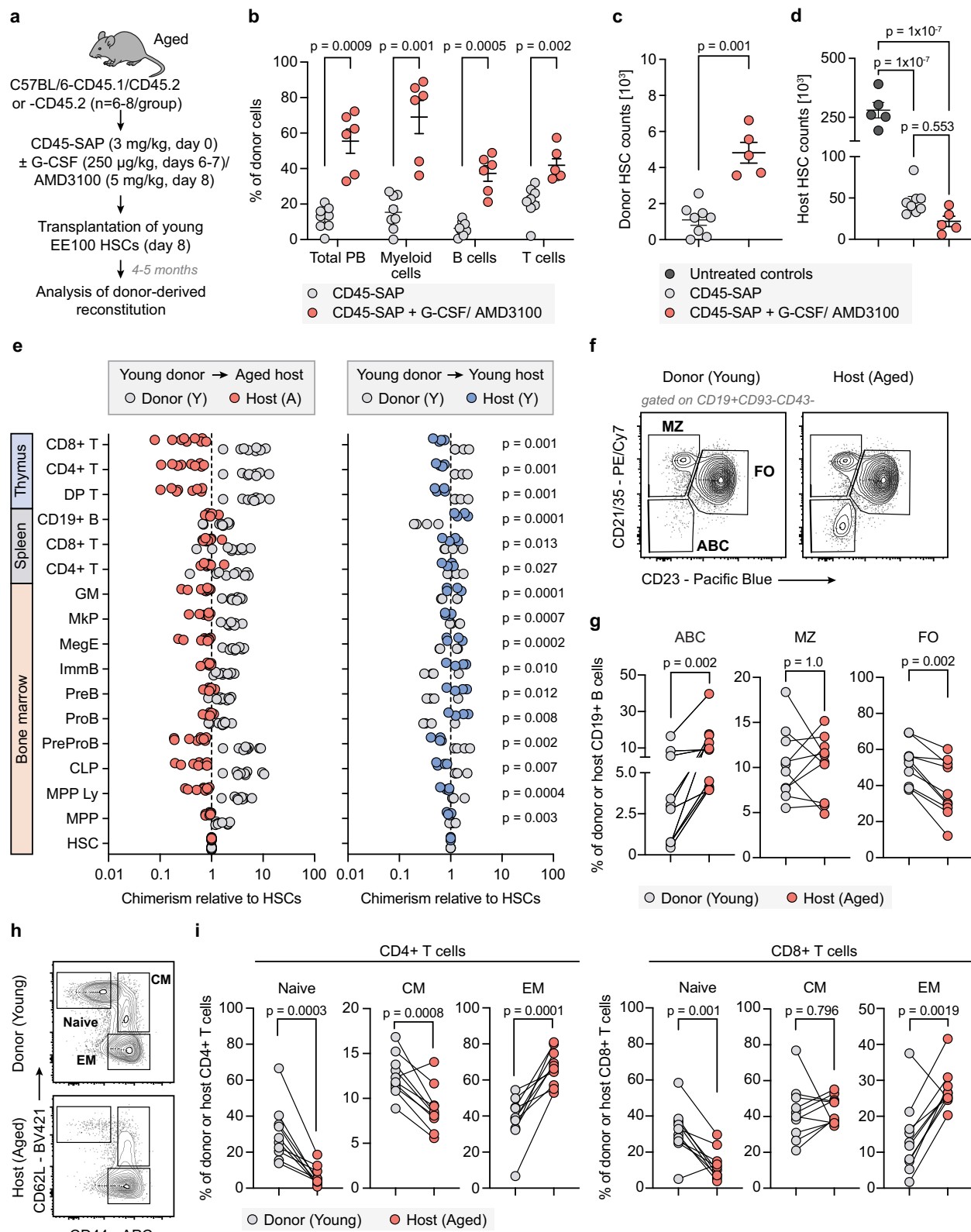

reconstitution associated with a pronounced bias toward myeloid cell generation (Supplementary Fig. 3b). BM HSC analysis further revealed nearly four times more recoverable donor HSCs in mice receiving the combined treatment than in those with CD45-SAP conditioning only (Fig. 4c). Consistent with earlier data (Fig. 3b), CD45-SAP significantly reduced host HSC levels, with further reductions observed in mobilized recipients (Fig. 4d).

We next examined the characteristics of young HSC-derived hematopoiesis and its interplay with the host's aged-derived hematopoiesis. We employed multi-parameter flow cytometry to stage hematopoiesis in the BM and, given the notable impact of aging on lymphopoiesis[14], conducted detailed examinations of B and T cell compartments in the spleen and thymus. Our analysis also included a small cohort of young recipients undergoing the same transplant procedure.

**Fig. 4 | Young HSCs support youthful hematopoietic characteristics upon transplantation into aged recipients. a** Experimental design. Aged (16 months) mice were treated with CD45-SAP (*n* = 8) or CD45-SAP with G-CSF/AMD3100 (*n* = 6) and transplanted with ex vivo expanded young HSCs (EE100 cells/mouse). **b** Young HSC-derived PB reconstitution in aged hosts 18 weeks after transplantation. **c–d** Absolute numbers of donor (**c**) and host (**d**) HSCs in aged recipients treated with CD45-SAP with or without G-CSF/AMD3100. In (**d**), HSC numbers in age-matched untreated controls were assessed for comparison (*n* = 5). In (**b–d**), gray, red, and dark gray circles represent CD45-SAP-, CD45-SAP/G-CSF/AMD3100-treated and untreated groups, respectively. **e** Chimerism levels in indicated hematopoietic cell subsets relative to the donor (young)- or host (aged)-derived HSCs in aged recipients (*left*, *n* = 10). Young recipients were used for comparison (*right*, *n* = 4). *p* value for each comparison is presented. Gray, red, and blue circles represent young donor, aged host, and young host groups, respectively. **f** Representative flow cytometric profiles of splenic B cell subsets within donor (young) and host (aged) cell fractions. ABC—age-associated B cells; MZ—marginal zone B cells; FO—follicular *B* cells. **g** Frequency of splenic ABC, MZ, and FO cells within donor (young)- and host (aged) CD19+ cell fractions (*n* = 10). **h** Representative flow cytometric profiles of splenic CD4 + T cell subsets within donor (young) and host (aged) cell fractions. CM—central memory; EM—effector memory. **i** Frequency of splenic naive, CM, and EM T cells within donor (young)- and host (aged) CD4+ (*left*) and CD8+ (*right*) cell fractions (*n* = 10). In (**g**) and (**i**), gray and red circles indicate donor (young) and host (aged) groups, respectively. In (**b–e**), (**g**), and (**i**), points indicate values for individual mice. In (**b–d**), error bars denote mean ± SEM. Statistical significance was determined by unpaired two-sided *t* test with Welch correction in (**b–c**) and (**e**), one-way ANOVA with a Tukey test in (**d**), two-sided Wilcoxon matched-pairs signed rank test in (**g**) and paired two-sided *t* test in (**i**). Source data are provided as a Source Data file.

To explore the relationship between transplanted and host HSCs and their differentiated progeny, we compared the chimerism levels of HSC progeny to those of BM HSCs. This consistently demonstrated that chimerism in progeny from young HSCs was significantly higher than in host-derived cells in aged recipients, but less pronounced in young hosts (Fig. 4e).

For the lymphoid lineages, the contribution to the early stages of differentiation (MPP Ly and CLPs) in aged recipients was almost five times higher than that of the HSCs themselves (Fig. 4e). Further examination of the B cell lineage revealed a slightly lower chimerism at the later B cell progenitor stages, although differentiation into these stages was still considerable higher than that observed for the age-derived cells. By contrast, early B cell progenitors in young recipients were predominantly host-derived (Fig. 4e).

Examination of the thymus revealed higher chimerism across all evaluated T cell subsets compared to cells derived from the host. Importantly, this difference was significantly more pronounced in aged recipients relative to their younger counterparts (Fig. 4e). Additional assessment of splenic B and T cells showed that while the overall chimerism from the young donor was greater than that for BM HSCs, these levels were generally not as high as those observed in the primary hematopoietic organs associated with these lineages (Fig. 4e).

Aging has been reported to correlate with the accumulation of a specific B cell subset known as age-associated B cells (ABCs)[29,30]. Therefore, we examined the presence and origins of ABCs, in conjunction with traditional follicular and marginal zone (MZ) B cells analysis (Fig. 4f). Although a small proportion of ABCs originated from the young donor, the overwhelming majority of these cells were host-derived (Fig. 4f, g). Conversely, donor cells effectively generated follicular B cells, while their contribution to the MZ *B* cell compartment was similar to the host-derived cells (Fig. 4f, g).

Next, we examined the distribution of more mature T cell subsets within the young to aged chimeras. Aging associates with an increased frequency of both central and effector memory cells, alongside a corresponding decrease in naive T cells[14]. Our analysis revealed that naive CD4 and CD8 T cells from host/aged-derived cells accounted for only about 5 and 15%, respectively, in stark contrast to the roughly 40% chimerism in both subsets derived from young HSCs (Fig. 4h, i). This underscores the potential to significantly boost the production of naive CD4 and CD8 T cells in aged mice.

A recent study suggested that selectively depleting aged HSCs through antibody-mediated targeting could mitigate age-related lymphoid deficiencies and potentially enhance immune function in the elderly[31]. Using CD45-SAP to deplete hematopoietic cells (Fig. 1), we compared mature *B* and *T* cell subsets in aged mice with those in unmanipulated aged controls (Supplementary Fig. 3d). While the absolute numbers of these cell populations remained lower than in young mice, we still observed reductions in host memory CD4 and CD8 T cells and follicular B cells compared to aged steady-state controls (Supplementary Fig. 3e-h).

Overall, these findings indicate that depleting host HSCs and transplanting young donor cells not only endows their progeny with youthful hematopoietic traits in aged recipients but also that the CD45-SAP treatment contributes to reverting the composition of the aged host's adaptive immune cells to a more youthful-like state.

## Molecular stability in early young donor-derived lymphoid progenitors exposed to aging BM

Age-related decline in lymphopoiesis can be linked to reduced production of early lymphoid progenitors, including MPP Ly[15,16]. Because this subset was effectively regenerated from young donor cells in aged recipients (Fig. 4e), our subsequent analysis examined the molecular features of these cells. For this, we performed RNA-sequencing of donor- and host MPP Ly cells from young and aged recipients (Supplementary Fig. 4a and b). Unexpectedly, principal component analysis failed to separate between donor and host MPP Ly across age groups (Supplementary Fig. 4c) and differential gene expression analysis revealed only 17 upregulated genes in donor MPP Ly from aged mice (Supplementary Fig. 4d). Similarly, analysis of host cells identified merely 16 upregulated genes upon aging (Supplementary Fig. 4e), without association to any MSigDB pathway (Supplementary Fig. 4f). Additional pairwise comparisons of donor and host cells within individual recipients yielded more differentially expressed genes (DEGs) (Supplementary Fig. 4g–h); however, lineage affiliation based on these genes revealed no notable differences among the groups (Supplementary Fig. 4i).

Together, these results revealed no significant differences in the transcriptomic signatures of donor-derived MPP Ly cells, even when exposed to an aging environment. This corroborates our functional data, demonstrating that transplantation of young HSCs effectively regenerates hematopoiesis with youthful characteristics in aged hosts (Fig. 4).

## Non-genotoxic BM conditioning followed by transplantation mitigates disease progression in a mouse model of myelodysplastic syndrome

The capacity to successfully reconstitute aged recipients (Fig. 4) established a basis for exploring this treatment in other compromised environments, such as those associated with age-related hematological disorders. Therefore, in our final experiments, we examined how CD45-SAP conditioning and HSC transplantation affect the development of age-associated hematological malignancies in the NUP98-HOXD13 (NHD13^tg) transgenic mouse model, which predisposes to myelodysplastic syndrome (MDS) and acute leukemia[32].

Our experimental layout entailed monitoring the disease evolution in NHD13^tg mice for their entire lifespan (up to 24 months, n = 20). This group was juxtaposed against a cohort of NHD13^tg mice that underwent CD45-SAP conditioning and transplantation with $10^7$ WT BM cells when they were 2 months old (n = 9). The choice of 2 months as the treatment age was based on two considerations: first, while the

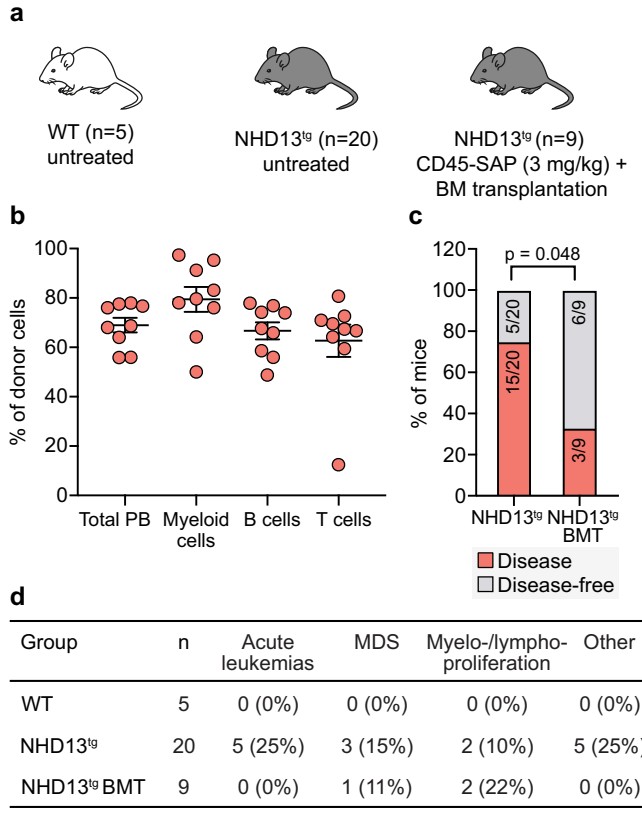

**Fig. 5 | Non-genotoxic BM conditioning followed by transplantation mitigates disease progression in a mouse model of myelodysplastic syndrome. (a)** Experimental design for Fig. 5b–d. Young (2 months) NHD13 transgenic (tg) mice were treated with CD45-SAP and transplanted with $10 \times 10^6$ BM cells derived from C57BL/6-CD45.1 mice ($n = 9$). Untreated NHD13$^{tg}$ ($n = 20$) and wild-type (WT, $n = 5$) littermates served as controls. Mice were monitored for disease development and donor-derived chimerism for up to 24 months of age. **b** Donor-derived reconstitution in indicated PB lineages 16 weeks after transplantation. The data for males and females are pooled and presented, with points indicating values for individual mice. Shown is the mean ± SEM. **c** Frequency of mice that developed hematological disease (red) or remained hematological disease-free (gray). Statistical significance was determined using a two-sided Fisher's exact test. **d** Disease type of individual mice. Values in parentheses indicate frequencies. Source data are provided as a Source Data file.

mice are still considered juvenile, the NHD13$^{tg}$ model begins to exhibit segmental aspects of aging-associated diseases already at a young chronological age. Additionally, we hypothesized that non-genotoxic transplantation could serve as a prophylactic strategy to mitigate disease progression. We also incorporated a small group (n = 5) of WT littermate mice as a control (Fig. 5a).

Mice subjected to CD45-SAP conditioning and transplantation exhibited high-level donor multilineage reconstitution four months post-transplantation (Fig. 5b). None of the aged WT mice developed hematological malignancies during the 2-year observation period. In contrast, many untreated NHD13$^{tg}$ mice began showing signs of diverse hematological diseases, including both myelo- and lymphoproliferative disorders and acute myeloid and T-cell leukemia, after six months of age. Although not every case could be conclusively diagnosed, many of the unclassified conditions were associated with pronounced thymic hyperplasia.

Overall, the incidence of disease in transplanted mice was significantly lower compared to their non-transplanted counterparts. Among the NHD13$^{tg}$ mice, 75% (15 out of 20) developed hematological malignancies, compared to only 33% (three out of 9) of those receiving WT cell transplants (Fig. 5c). Furthermore, while 25% (five out of 20) of

NHD13$^{tg}$ mice developed acute leukemia, none of the transplanted mice did (Fig. 5d).

## Discussion

In this work, we explored non-genotoxic BM conditioning as a regimen for providing aged recipients with HSCs from younger donors. Traditional conditioning commonly employs varied levels of TBI. However, TBI associates with systemic side effects that are poorly tolerated by aged recipients[33], emphasizing the need for alternative conditioning approaches. The CD45-SAP/G-CSF-based regimen optimized in this study offers specificity, reduced toxicity, and lower tissue inflammation. The use of CD45-SAP in aged recipients further holds the potential to eliminate age-associated senescent cells, which otherwise contribute to tissue decline and potentially reduce transplantation success. Unlike previous strategies requiring multiple rounds of G-CSF/AMD3100-based mobilization[28], our approach uses a single round of conditioning, further minimizing recipient suffering and avoiding the risk of developing anti-G-CSF antibodies[34].

While BMT is a well-established clinical procedure, many aspects concerning successful HSC reconstitution remain elusive. The recognition that HSCs inhabit niches crucial for regulating their function has led to a central hypothesis suggesting the necessity for these niches to be available for the seeding of transplanted HSCs[35]. Moreover, it has been proposed that a scarcity of such niches could limit HSC engraftment. Consistent with this notion, a temporary mobilization combined with transplantation at the mobilization peak has been recently proposed to enhance reconstitution in otherwise unconditioned hosts[8]. Interestingly, additional suggestions propose that repeated mobilization and transplantation may not only result in significant hematopoiesis derived from younger sources but also extend lifespan[28]. In line with the effectiveness of this conditioning method, we confirmed that combining it with CD45-SAP significantly enhances reconstitution in aged hosts. Still, the concept of niche recycling could be more complex, given recent findings that suggest it is possible to efficiently reconstitute unconditioned hosts with massive numbers of infused HSCs[10]. However, the requirement to achieve high-level engraftment - necessitating HSCs from nearly 300 mice - poses severe experimental limitations[10]. Instead, we capitalized on recent breakthroughs in HSC ex vivo expansion[11]. This approach facilitated donor chimerism in young unconditioned hosts, which could be significantly boosted in combination with CD45-SAP-mediated conditioning.

As part of our work, we tried to reassess the concept of niche recycling by comparing the contribution from one dose of HSCs to the same dose infused over five separate occasions. This resulted in high overall reconstitution levels that, however, were not further elevated in the setting of repetitive transplantation. This contrasts with previous studies suggesting increased graft contribution with constantly vacated niches[9]. We attribute this deviation to the need for extremely high quantities of HSCs, not previously attainable, to saturate the available niches of an unconditioned recipient[10].

One of our key initial findings was that aged mice presented significant barriers to effective engraftment, which we attribute, at least in part, to the less effective conditioning and insufficient depletion of endogenous aged HSCs. This may be due to the initially higher cell counts in aged mice, which could lead to a greater peripheral uptake of CD45-SAP. Additionally, systemic changes associated with aging, such as decreased hepatic and renal function and changes in the body composition, may further influence immunotoxin pharmacokinetics, mode of action, and its bioavailability in the BM. By addressing these limitations through the combination of two different non-genotoxic conditioning techniques and increasing donor HSC doses, we successfully achieved substantial hematopoiesis from transplanted young HSCs in aged recipients.

It has been hypothesized that aging is associated with changes in BM niches, potentially impacting HSC function[36]. Additionally, aging may contribute to other systemic changes that could compromise HSC functionality[37,38]. If such age-related alterations are significant, introducing young HSCs into aged recipients might not effectively restore youthful hematopoietic function. Nonetheless, aged individuals still possess HSC clones capable of contributing to multilineage hematopoiesis when transplanted into younger hosts[15]. This suggests that aging may limit the contribution from these clones, as studies have shown that eliminating senescent cells[17] or specifically targeting functionally aged HSCs[31] can enhance host HSC function. Our findings further support this by demonstrating that young HSCs transplanted into aged hosts behave differently from the aged host HSCs, including displaying a more robust lymphoid output.

Age-related chronic inflammation, similar to TBI-induced inflammation, has been suggested to influence HSC behavior at least partly by altering their proliferative state[39]. By using CTV-based proliferation tracking following transplantation, we revised this concept. First, we observed that CD45-SAP treatment, like TBI[40], induces extensive proliferation of transplanted cells regardless of the recipient's age. This was accompanied by a loss of functional potential. Second, unlike CD45-SAP treatment, unconditioned transplantation preserved a fraction of undivided HSCs, with a trend for a higher proportion of these cells in aged recipients. Intriguingly, our secondary transplantations revealed enhanced multilineage reconstitution from these undivided cells, perhaps suggesting that quiescence of transplanted HSCs may protect their functional capacity during aging. However, when interpreting these results, it must be considered that our CTV-tracking experiments failed to assess the potential re-entry of transplanted HSCs into quiescence.

Aging is widely recognized to result in dampened adaptive immunity and, in particular, neo-responses[41], making the restoration of robust lymphopoiesis in aged hosts a primary objective. We illustrate that young HSCs can facilitate highly efficient BM B lymphopoiesis in aged recipients, challenging the idea that the aged environment inherently exerts negative influences on this differentiation pathway[42]. Similarly, we observed a notable increase in thymic seeding and/or expansion from the transplanted young grafts, which correlated with elevated numbers of naive donor-derived peripheral T cells. These findings suggest that introducing young HSCs can enhance thymic function in older recipients despite the pronounced thymic involution with age.

Lastly, we aimed to assess whether the non-genotoxic transplantation could serve as a prophylactic treatment to slow or prevent the progression of hematological diseases. We envision this strategy as particularly applicable to disorders associated with germline mutations in genes, such as *SAMD9*, *SAMD9L*, *DDX51*, or *GATA2*, which can often be identified before the onset of disease[43]. By leveraging the NHD13[tg] mouse model, we discovered that the introduction of young HSCs into asymptomatic NHD13[tg] mice led to a noteworthy decrease in hematological disease occurrence and, strikingly, to a complete inhibition of acute leukemia development. As at least some host hematopoiesis remains after CD45-SAP conditioning and transplantation, we interpret these results to illustrate that young and genetically intact HSCs can function as a tumor suppressor, perhaps through a cell competition mechanism reminiscent of the Scribble mutation in *Drosophila*[44].

In conclusion, we here tried to shed light on the potential of non-genotoxic BM conditioning as an effective strategy to introduce young HSCs into aged hosts. Crucially, we have demonstrated that young HSCs can enhance adaptive immune cell generation, even within an aged milieu, and we highlight the potential of combined non-genotoxic conditioning with HSC transplantation to hinder the progression of age-related hematological disorders.

Our study focused on enhancing hematopoiesis in the elderly through non-genotoxic transplantation, but several limitations must be acknowledged. First, while we characterized the effects of the CD45-SAP conditioning regimen on transplanted young HSCs in aged recipients, their extensive self-renewal capacity was not fully explored. We deemed this less relevant given that our primary goal was to improve hematopoiesis within the lifespan of the transplanted hosts rather than to evaluate long-term self-renewal beyond this timeframe.

Second, while we demonstrated the capacity of young donor HSCs to regenerate the adaptive immune compartment, we did not perform functional assessments of immune rejuvenation, such as the ability to mount responses to infections or vaccinations, leaving this aspect of therapeutic potential unvalidated.

Additionally, our transcriptomic profiling of MPP Ly cells post-transplantation provided insights into molecular changes but lacked comparisons with steady-state, untreated controls. This limits our ability to draw definitive conclusions about the mechanisms underlying MPP Ly aging or their modulation by treatment. A limitation of our study is that some experiments were conducted with relatively low sample sizes. This is primarily due to the challenges of working with aged mice, which are not easily available, and the nature of certain transplantation experiments, where obtaining sufficient cell numbers was a limiting factor.

## Methods

### Ethical statement
This study complies with ethical regulations at Lund University. Animal research was approved by the Malmö-Lund Animal Experimentation Ethics Committee (Malmö - Lunds djurförsöksetiska nämnd).

### Mice
Animal experiments were performed according to the ethical permit protocols M186-15 and 16468/2020, approved by the Malmö-Lund Animal Experimentation Ethics Committee. All experiments involved young (2-4 months) and aged (16-20 months) C57BL/6-CD45.2, C57BL/6-CD45.1 or F1 C57BL/6-CD45.1/CD45.2 mice (*Mus musculus*) obtained from Jackson Laboratory, Janvier Labs, Taconic Bioscience, or generated in-house. NHD13[tg] mice[32] (RRID: IMSR JAX:010505) were obtained from Jackson Laboratory. All analyses were performed on female mice, except in Figs. 2f, 5, which involved both males and females. The inclusion of both sexes in Fig. 5 aimed to address potential sex-related differences in the disease latency and treatment outcome. The decision to use only females in most of the studies was based on practical considerations that co-housed males more frequently exhibit aggression and fighting, which could interfere with longitudinal experiments. For transplantation of tdTomato+ HSCs (Fig. 1 and Supplementary Fig. 1), cells were isolated from *Fgd5*[CreERT2/+]; *Rosa26*[LSL-tdTomato/+] mice, generated by crossing Fgd5-ZsGreen-2A-CreERT2 mice[45] (RRID: IMSR JAX:027789) to Rosa26-LoxP-STOP-LoxP-tdTomato (RRID: IMSR JAX:007905) mice. tdTomato-labeling in *Fgd5*-expressing HSCs was induced with tamoxifen (Sigma Aldrich, 10 mg/ml, resuspended in peanut oil) by intraperitoneal injections at 50 mg/kg[15]. Mice were housed in the Animal Facility at the Biomedical Center of Lund University in enriched environment conditions, with a 12 h light-dark cycle, a temperature of 22 °C, 55% relative humidity, and *ad libitum* access to food and water.

### Animal procedures
**Bleeding and isolation of BM, thymus, and spleen cells.** PB was collected from the tail vein into EDTA-coated tubes (Sarstedt) or 2% (v/v) FBS/PBS with heparin (Leo Pharma, 5000 IE/ml diluted 1:500). Complete blood count was determined using Sysmex KX-21N and XQ-320 analyzers. For BM cell isolation, mice were euthanized by cervical dislocation, and femurs, tibias and hip bones were collected from both hind legs. Bones were crushed in ice-cold 2% (v/v) FBS/PBS. For isolation of thymus

and spleen cells, organs were dissociated using a plunger and 70 μm strainer in ice-cold 2% (v/v) FBS/PBS. Single-cell suspensions were centrifuged at 400 g for 10 min and filtered through 70 μm cell strainers prior to sample processing.

**Immunotoxin preparation and conditioning.** The CD45-SAP immunotoxin was prepared as previously described[7]. Biotinylated anti-CD45.2 antibodies (clone 104, Sony Biotechnology) were mixed with streptavidin-saporin conjugate (Advanced Targeting Systems, cat. no. IT-27, Lot #132-178 and #201-151) at a 1:1 molar ratio. In all experiments, CD45-SAP was administered at 3 mg/kg 8 days before analysis or transplantation, except in cohorts of young and aged animals in Fig. 4e and Supplementary Figs. 3 and 4, which received CD45-SAP at 60 μg/mouse 20 weeks after transplantation. For CD4-SAP and CD8-SAP treatment, biotinylated anti-CD4 (clone GK1.5, Sony Biotechnology) or anti-CD8a (clone 53-6.7, Sony Biotechnology) was combined with streptavidin-saporin conjugate and injected at 0.5 mg/kg 2–3 days before transplantation. All immunotoxins were diluted in PBS and administered intravenously. B cell depletion was performed as previously reported[46] by intraperitoneal injection of rat anti-mouse CD19 and CD22 (clones 1D3 and CY34.1, respectively; BioXCell) and rat anti-mouse B220 (clone RA3-6B2; eBioscience) antibodies at 150 μg/mouse. After 48 h, mice received intraperitoneal injections of anti-rat kappa light chain (clone MAR18.5; BioXCell) at 150 μg/mouse.

For TBI, mice were sublethally (200 cGy) or lethally (950 cGy) irradiated one day or four hours before transplantation, respectively. All mice received antibiotic prophylaxis (Ciprofloxacin, HEXAL, 125 mg/l in drinking water) for two weeks following conditioning.

**G-CSF/AMD3100 mobilization.** Mice received subcutaneous injections of recombinant human G-CSF (Zarzio 48 MU/0.5 ml, Sandoz GmbH) at 125 μg/kg every 12 h for two days. Eighteen hours after the last G-CSF injection, mice received AMD3100 at 5 mg/kg in PBS (Sigma, cat. no. A5602-5MG). One hour following the AMD3100 injection, mice were transplanted with ex vivo expanded HSCs.

**Transplantation.** All transplantations were performed through tail vein injection. See Supplementary Table 1 and corresponding figure legends for detailed experimental descriptions. For HSPC transplantation in Fig. 1 and Supplementary Fig. 1, tdTomato+ HSCs were isolated by FACS from tamoxifen-induced $Fgd5^{CreERT2/+}$; $Rosa26^{LSL-Tomato/+}$ mice and 500 cells were injected alongside 500,000 BM cells from C57BL/6-CD45.1 mice into each young and aged recipient. For transplantation of ex vivo expanded HSCs, cultured cells were collected, washed, and filtered/FACS-purified prior to transplantation. In all experiments, cells from cultures were pooled, and each recipient received an equal portion of the same input HSCs. For secondary transplantation (Fig. 2f), $3 \times 10^6$ BM cells from primary recipients were either non-competitively transplanted or mixed with whole BM-derived from C57BL/6-CD45.1/CD45.2 mice at a 1:1 ratio ($3 \times 10^6$ cells of respective fraction per mouse) and transplanted into TBI-treated C57BL/6-CD45.1/CD45.2 secondary hosts. Cells from primary recipients were pooled before transplantation. For in vivo proliferation tracking, cultured cells and CD4-enriched splenocytes isolated from C57BL/6-CD45.1 or C57BL/6-CD45.2 mice were labeled with CTV dye as previously described[12]. EE100 HSCs were co-transplanted with $2 \times 10^6$ splenocytes into each C57BL/6-CD45.2 or C57BL/6-CD45.1/CD45.2 unconditioned or CD45-SAP treated recipient.

**Disease classification in NHD13$^{tg}$ mice.** Classification of hematopoietic diseases was based on the following criteria: Myelodysplastic syndrome was characterized by WBC < 5 and anemia and/or thrombocytopenia. Lympho- and myeloproliferation were identified by 50 > WBC > 20; acute myeloid and lymphoid leukemia were defined by a WBC > 50 and their respective myeloid- or lymphoid-lineage assignments as determined by flow cytometry.

### Ex vivo HSC expansion

Ex vivo HSC expansion was performed as described before[11]. Briefly, 96-well flat-bottom plates were coated with 100 ng/ml fibronectin (Sigma, cat. no. F0895) for at least 1 h. HSCs (Lineage-cKIT+SCA-1 + CD48-CD150+CD201$^{high}$) were isolated from young mice and cultured for 21 days in Ham's F12 medium (Gibco, cat. no. 11765054) supplemented with 1% insulin−transferrin−selenium−ethanolamine (Gibco, cat. no. 51500056), 10 mM HEPES (Gibco, cat. no. 15630080), 1% penicillin/streptomycin/glutamine (Gibco, Cat. no. 10378-016), 0.1% PVA (87−89%-hydrolyzed, Sigma, cat. no. 363081), 10 ng/ml mouse SCF (Peprotech, cat. no. AF-20503) and 100 ng/ml mouse THPO (Peprotech, cat. no. AF-31514) at 37 °C with 5% $CO_2$ and 20% $O_2$. Media changes were performed every 2 days starting from day 5 after sorting. Cells were split when reaching 80-90% confluency.

### Flow cytometric analysis and FACS sorting

Fluorescently labeled and biotinylated antibodies used in this study are listed in Supplementary Table 2. Stainings that included Brilliant Violet-conjugated reagents were supplemented with 10% Brilliant Stain Plus Buffer (BD, cat. no. 566385). The cells were stained in 2% (v/v) FBS/PBS for 30 min at 4 °C in the dark, unless otherwise stated. For PB analysis, erythrocytes were sedimented with 1% Dextran T500 (Sigma-Aldrich) at 37 °C for 30 min. The remaining erythrocytes were then lysed with ammonium chloride solution (STEMCELL Technologies, cat. no. 07800) for 3 min at room temperature. Cells were stained in 2% (v/v) FBS/PBS with 2 mM EDTA (VWR) and antibodies against TER119, CD19, CD11b, Gr1, NK1.1, and CD3.

HSPC analysis was performed on whole BM or cKIT-enriched cells. For cKIT enrichment, BM cells were stained with anti-cKIT-APC or anti-cKIT-APCeFluor780 antibody, followed by incubation with anti-APC MicroBeads (1:20, Miltenyi Biotec, cat no. 130-090-855) for 30 min. Magnetic separation was performed using LS or MS columns and a manual separator, according to the manufacturer's instructions (Miltenyi Biotec, cat. no. 130-042-401).

In all stainings, except for myelo-erythroid cell subsets, cells were pre-incubated with Fc-block (1:50, BioXCell, cat no. BE0307) for 15 min prior to antibody staining. For HSPC analysis, cells were stained with antibodies against lineage markers (B220, Gr1, TER119, NK1.1, CD3), SCA-1, cKIT, CD150, CD48, CD201, and in some experiments also against CD135 and CD127. Myelo-erythroid progenitors were identified using lineage markers (B220, Gr1, TER119, NK1.1, CD3), SCA-1, cKIT, CD150, CD105, CD16/32, and CD41. B cell progenitors were identified using lineage markers (Gr1, TER119, NK1.1, CD3), CD19, B220, IgM, CD93, CD43, and cKIT.

For thymocyte analysis, cells were stained with antibodies against CD19, B220, CD3, CD4, and CD8. For mature *B* and *T* cells, splenocytes were stained with antibodies against CD19, CD93, CD23, CD21/35, CD43, CD11b, and CD11c for *B* cell lineage or Gr1, CD4, CD8, CD3, CD44, and CD62L for T cell lineage. In transplantation experiments, all staining panels included antibodies against CD45.1 and CD45.2 to monitor chimerism levels.

HSC isolation by FACS was performed on cKIT-enriched BM cells stained with the HSPC antibody cocktail described above. For isolation of cultured HSCs, cells were stained against lineage markers (Fcer1a, B220, Gr1, TER119, NK1.1, CD3) and CD201. Prior to analysis or cell sorting, cells were filtered and incubated with propidium iodide (1:1000, Invitrogen) to exclude dead cells.

All flow cytometry and FACS experiments were performed at Lund Stem Cell Center FACS Core Facility (Lund University). Flow cytometric analyses were conducted on LSRFortessa or Fortessa-X20, and cell sorting was performed on FACSAria III or FACSSymphony S6 instruments (Becton Dickinson). Data was analyzed using FlowJo v.10.8.1

(Treestar). Representative flow cytometry plots and gating strategies for all evaluated cell populations are included in Supplementary Figs. 5 and 6.

## RNA-sequencing and bioinformatic analysis

Young (3 months) and aged (16 months) G-CSF/AMD3100-treated mice were transplanted with ex vivo expanded HSCs. After 20 weeks, mice were treated with CD45-SAP at 20 µg/mouse to deplete host BM cells, and hematopoietic reconstitution was monitored for additional 16 weeks. At the endpoint, MPP Ly cells (Lineage-cKIT+SCA-1 + IL7Ra-CD135 + CD150-) were isolated from the donor (young) and host (young or aged) BM fractions for bulk RNA-sequencing analysis. Cell isolation was performed in two batches, with each batch consisting of donor and host cells from both young and aged recipients. cDNA library preparation and sequencing were performed at Single Cell Discoveries (Utrecht, the Netherlands) using a modified CEL-Seq2 protocol. Briefly, 200 MPP Ly cells were isolated by FACS, and total RNA was extracted using TRIzol reagent (Invitrogen, cat. no. 15596026). mRNA was reverse transcribed, barcoded, pooled, and amplified with an in vitro transcription. The resulting RNA was fragmented and used to generate cDNA sequencing libraries with Truseq Small RNA adapters (Illumina). Libraries were paired-end sequenced on a NovaSeq X Plus instrument using a 10B Reagent kit (100 cycles) and the following read configuration: R1 = 26 cycles, i7 = 6 cycles, R2 = 60 cycles (Illumina).

After sequencing, data were demultiplexed and mapped to the mouse GRCm38 reference genome using STARsolo v.2.7.10b software. Reads mapping to multiple locations were discarded. Subsequent analyses were performed in R v.4.2.1, including normalization and differential gene expression using DESeq2 v.1.36.0 package[47]. PCA plot was generated using the *vst()* function of DESeq2. DEGs were identified based on an adjusted *p* value < 0.1. Information about the batch and sample pairing were included as covariates to account for technical variation and within-subject variability. Venn diagrams were generated using Venny v.2.0.2 (https://bioinfogp.cnb.csic.es/tools/venny/index2.0.2.html), and pathway analysis was conducted using Enrichr (https://maayanlab.cloud/Enrichr/). Lineage affiliations based on identified DEGs were computed using the Cell Radar tool (https://karlssong.github.io/cellradar/).

## Statistical analysis

Data were analyzed and visualized using Microsoft Excel v.16.66.1 (Microsoft), Prism 9 v.9.3.1 (GraphPad), and R v.4.2.1 software. All illustrations, including those depicting experimental setups, were created by the authors using Adobe Illustrator (v26.4.1). Results are presented as mean ± SEM unless otherwise stated. Experiments were repeated as indicated in figure legends, with n denoting the number of independent biological repeats. Two-group comparison with normally distributed data employed a two-tailed *t*-test with Welch correction, while not normally distributed data were analyzed using the Mann-Whitney U test. Statistical analyses were unpaired, except in Fig. 4g, i, which involved paired comparisons. Multiple group comparisons were assessed by one-way ANOVA with post hoc Tukey correction. Specific tests used are indicated in the corresponding figure legends.

## Reporting summary

Further information on research design is available in the Nature Portfolio Reporting Summary linked to this article.

## Data availability

The Bulk RNA-sequencing data generated in this study have been deposited in the GEO database under accession code GSE267079. [https://www.ncbi.nlm.nih.gov/geo/query/acc.cgi?acc=GSE267079]. Source data are provided with this paper.

## Code availability

No original code is reported in this manuscript.

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

## Acknowledgements
We thank Single Cell Discoveries for their assistance with bulk RNA-sequencing and S. Soneji for suggestions on bioinformatic analysis. We acknowledge the expertise and assistance of the staff at the Lund University Animal Facility and Lund Stem Cell Center FACS Facility. The study was generously supported by the Tobias Foundation (Tobias Prize to D.B.), the Swedish Research Council grant 2022-00932 (D.B.), the Swedish Cancer Society 211470Pj (D.B.), the Swedish Pediatric Leukemia Foundation PR2022-0091 (D.B.) and by the Royal Physiographic Society of Lund foundation 42335 and 43250 (A.K-C.) and 42331 and 43043 (Q. Z.).

## Author contributions
Conceptualization: A.K.-C. and D.B. Methodology: A.K.-C. and Q.Z. Investigation: A.K.-C., Q.Z., and S.K. Visualization: A.K.-C. and D.B. Funding acquisition: A.K.-C., Q.Z., and D.B. Project administration: A.K.-C. and D.B. Supervision: D.B. Writing—original draft: A.K.-C. and D.B. Writing—review & editing: A.K.-C. and D.B.

## Funding

## Competing interests
The authors declare no competing interests.
