## [Transparent Peer Review file · Nature Communications]

A Non-Genotoxic Stem Cell Therapy Boosts Lymphopoiesis and Averts Age-Related Blood Diseases in Mice

Corresponding Author: Professor David Bryder

Version 0:

Reviewer comments:

Reviewer #1

(Remarks to the Author)

Aging is associated with a decline in hematopoietic fitness and increased risk of hematological malignancies, in part arising from defects within hematopoietic stem cells (HSC). These HSC defects manifest clinically as deficient lymphopoiesis and a decrease in adaptive immunity. In the submitted manuscript entitled "A Non-Invasive Stem Cell Therapy Boosts Lymphopoiesis and Averts Age-Related Blood Diseases in Mice" by Konturek-Ciesla et al, the authors develop and optimize a non-myeloablative HSC transplantation protocol wherein HSC from young mice are adoptively transferred into aged mice that are conditioned with a combined CD45 depletion (CD45-SAP) and G-CSF mobilization strategy. Utilizing this protocol, the authors demonstrate successful engraftment of young HSC into aged recipients. They demonstrate an improvement in phenotypic lymphopoiesis parameters by flow cytometry in aged mice. Furthermore, they demonstrate the clinical utility of this protocol by transplanting wild type HSC into a myelodysplasia mouse model (2 month-old NHD13 mice) and tracking the incidence of blood disorders over a course of two years.

A handful of studies in the past decade have demonstrated alternative strategies to achieve successful HSC engraftment without utilizing myeloablative (chemotherapy/irradiation) conditioning regimens. However, the majority of these studies focused on young recipients. It is generally accepted that achieving successful engraftment in aged recipients is significantly more challenging than in younger recipients. Elderly recipients also experience higher toxicity with myeloablative regimens. Only one prior study has demonstrated successful engraftment in aged mice utilizing non-myeloablative regimens. Given the dearth of knowledge in non-myeloablative conditioning in the elderly demographic, the presented study will provide valuable information to scientists working in a wide ranging of disciplines including HSC transplantation, aging, and blood cancers, making this a good candidate to be considered for publication in Nature Communications. The experiments are generally well-designed and well-controlled. However, some statements and claims made by the authors are overstated and not supported by the data presented. The following concerns need to be addressed before being considered for publication.

Major Concerns:

- 1) The primary objective of the manuscript was to develop a safe and efficient conditioning protocol for transplanting young HSCs into an aged recipient. While secondary transplant data was provided for young conditioned recipients (Fig. 2F), it was not provided for the most critical group: aged conditioned recipients (either CD45-SAP alone or CD45-SAP plus G-CSF). In light of the impact of SAP on HSC cell cycle (Fig. 3D), secondary transplant data are crucial to evaluate the long-term safety and efficacy of this conditioning regimen on aged recipients, and their comparison with non-conditioned EE100/500 transplant recipients, steady state controls, and if feasible TBI transplant controls. If these data are not available, please include a Limitations section acknowledging that the impact of SAP procedure on HSC self-renewal in aged recipients remains to be evaluated.
- 2) One of the hallmarks of HSC aging is a myeloid-biased output at the expense of lymphopoiesis, both at steady state and following transplantation (Rossi et al PMID: 15967997). In Figure 4B, it appears that G-CSF causes an increase in myeloid-biased engraftment (higher myeloid chimerism as compared to total chimerism). Please include %myeloid cells, %B cells, and %T cells as a frequency of engrafted donor cells as described in PMID: 15967997 for Fig. 1F, 2D, 2H, 3F, and 4B. These reanalyzed data will provide valuable information as to whether the conditioning regimens (CD45-SAP, G-CSF etc) have any impact on myelopoiesis and lymphopoiesis capacity of transplanted HSC.
- 3) Anemia is a characteristic feature observed in aged C57BL6 mice. Extended data 1A shows that CD45-SAP causes anemia in the short term. Does this resolve over time? And does transplanting young HSC into aged recipients able to improve anemia (Fig. 4A)? Along these lines, complete blood counts in long term engrafted aged SAP/SAP-GCSF

recipients would significantly add to the manuscript.

- 4) The number of animals analyzed are very small for some experiments (eg., Fig.1A N=3-4/group; Fig.3A,B 2-3/group). While this is generally not a major concern when the effect sizes are large, it becomes problematic when the observed changes are small. For example, in Fig 3B, it appears that the number of recoverable donor HSC is lower in aged recipients when compared to young recipients in both unconditioned ($P=0.11$) and conditioned ($P=0.34$) settings. The authors interpret this data as "When analyzing CD45-SAP-conditioned young and aged hosts, we observed similar amounts of recoverable young donor HSCs (Fig. 3b). This demonstrated that compromised homing/engraftment in aged mice was unlikely to explain the inefficient reconstitution from young HSCs (Fig. 1)." This interpretation is problematic as the N is too small to make the conclusion that aging is associated with no homing defects. Moreover, this is not the ideal experiment to assess homing. The authors should either examine homing at short time points (eg 18 hours) or rephrase this statement to acknowledge these caveats.
- 5) Fig. 3C-D. CTV experiments clearly show that CD45-SAP conditioning causes a significant increase in HSC proliferation and a complete loss of HSC quiescence (20-30% quiescent in unconditioned vs 0% in CD45-SAP at 2-4 weeks). Were these experiments performed at the same time point for all groups (young, aged, conditioned, unconditioned) or some at 2 weeks and some at 4 weeks? Does the proliferation in SAP groups resolve by 6 weeks (Fig. 3E)? These are important to know because excessive proliferation can have deleterious impact on HSC self-renewal in the long-term.
- 6) In Fig.3F, please include total donor chimerism as reported for other transplant data.
- 7) For data in Fig. 4E-H and Extended Data 2, data are presented as 'normalized to young' which makes it difficult to evaluate the extent of lymphopoiesis restoration. Non-normalized absolute values for these parameters in steady state controls (young and aged unmanipulated controls), and conditioned aged recipients would be necessary to illustrate two points: age-related defects in lymphopoiesis and degree of reversal following stem cell therapy. These non-normalized data as bar graphs can be added in extended data.
- 8) The authors make the statement that "Taken together, these results establish that young HSCs can successfully engraft in an aged environment, which does not inherently harm HSCs or hinder their ability for proliferation or long-term multilineage reconstitution." While the authors clearly show that young HSCs can successfully engraft an aged environment, they do not have adequate data to support that the aged environment does not harm or hinder the transplanted young donor HSCs. Data in Fig. 3D contradicts this statement (% quiescent HSC in young vs aged unconditioned recipients). This difference in quiescence (young vs aged unconditioned), and the impact of SAP conditioning on HSC proliferation needs to be discussed in the manuscript.
- 9) In their RNA Seq analyses, the authors conclude that "Together, these results revealed no significant differences in the transcriptomic signatures of donor MPP Ly cells, even when exposed to an aging environment." However, the comparisons performed and controls included are insufficient to support this statement. If the conditioning regimen caused a large scale shift in the transcriptome, it could also explain the observed lack of significant gene expression changes across the 4 groups (all of which received conditioning). It would be worthwhile to perform comparisons of the 1) donor MPP (young) vs host MPP (aged), and 2) donor MPP (young) vs host MPP (young) which might provide some information as to whether the aging environment impacts MPP gene expression. Ideally, MPP from steady state (non-conditioned, non-transplanted) young and aged mice would need to be compared alongside. However, if these steady state data are not available, the interpretations should be performed in light of these caveats. Additionally, please include the time point after the transplant that the RNA Seq was performed and its experimental set-up in Extended Data Table 3.

Minor concerns:

- 1) The title is overly broad. It implies that the proposed stem cell therapy averts all aging related blood diseases in mice. It needs to be made specific to the data presented in the manuscript. HSC transplantation by its nature is an invasive procedure. Consider changing non-invasive to "non-chemo/irradiation based" based or "non-myeloablative" or "low-intensity conditioning".
- 2) The authors provide no functional readouts for improvements in lymphopoiesis (e.g vaccine challenge etc) or HSC self-renewal in aged conditioned recipients. Consequently, please tone down some of the statements including "not only survive but thrive in aged hosts, dramatically improving hematopoietic output and ameliorating age-compromised lymphopoiesis".
- 3) In the Discussion the authors state that "If such age-related alterations are significant, introducing young HSCs into aged recipients might not effectively restore youthful hematopoietic function...Firstly, we demonstrate that young HSCs transplanted into aged hosts exhibit behavior akin to their performance in a normal environment." To justify this statement, a comprehensive hematopoietic analysis needs to be performed (HSC self-renewal assays, comprehensive blood counts, HSC RNA Seq, functional assays for evaluating immunosenescence, evaluation of BM niches etc). If these data are not available, this statement should be toned down a bit.
- 4) A prior study demonstrated successful engraftment of young HSCs in aged mice (Reference 27, Guderyon, M.J., et al). In the Discussion, it would be useful to compare and contrast the CD45 SAP/GCSF protocol with this published study. Additionally, the relative advantages and disadvantages of CD45-SAP compared with other strategies cited in their references would add value in placing this protocol in the broader context.
- 5) The authors state that "We and others have previously established the efficacy of a polyvinyl alcohol (PVA)-based culture system in promoting murine HSC expansion". Please rephrase this to accurately credit Wilkinson et al for the discovery of PVA-based expansion. There should be no problem in having a second sentence stating that "we have validated this expansion platform in our prior study".
- 6) The authors state that "Conversely, the reconstitution levels were markedly decreased upon competitive transplantation, mirroring the reduction in HSCs derived from primary TBI-conditioned hosts (Fig. 2h)". It is not clear as to which reduction in HSCs are being referred to here.
- 7) Please include flow cytometry gating strategy for all flow cytometric analysis in Supplemental Data.

(Remarks to the Author)

The authors proposed a mouse model in which the CD45-SAP condition is used to treat hematologic diseases caused by aging HSCs by transplantation of young HSCs. The authors further developed their experimental results to ultimately treat MDS.

On the other hand, there is great concern about the authors' new finding.

The authors have a major concern about the new regulation, which, as they note in their interim summary, is that it does not offer anything new in comparison with previous reports.

As I understand it, they then go on to manipulate the transplant system in a complex way from Fig. 3, but I see no particularly surprising findings.

In addition, some concerns are noted below.

1) That's interesting that Old-HSC are not reduced by CD45-SAP compared to Young-HSC, but why are Old-HSC not reduced more than Young-HSC? Is it because of optimized conditions?

2) CD45-SAP as a model for aging hematopoiesis or MDS treatment

Auto's system seems inappropriate. What would happen in the experimental system of Allo's system (at least haplo-MHC mismatch)?

3) Related to 2), are CD45-SAP transplants rejected by donor cells in allo-HSC transplants?

4) Finally, the novelty (point) of the authors did not come across to me, as CD45-SAP is a paper that is quite dependent on and very similar to the previously reported paper by Palchaudhuri, R et al. If there is really a major novelty, it might be good to prepare a summary with figures that we see more often these days.

Minor comment)

It is confusing for the reader to tell from the figure which mouse is the donor and which is the recipient.

Reviewer #3

(Remarks to the Author)

This study primarily sought to investigate non-invasive BM conditioning as a regimen to promote the effective engraftment of young HSCs in aged recipients. It ultimately showed that the best results for aged mice were obtained when CD45-SAP was combined with a mobilization strategy to disrupt the niches. It also investigated whether a multiple smaller transplantations of HSCs was superior to a single large HSC transplantation in young mice, and finally showed that BM transplantation could reverse an inherent genetic predisposition to hematological malignancies in a transgenic mouse model. The data contain interesting findings. In particular, the analysis of how young BM reconstitutes in a young host vs an aged host, and also the qualitative analysis of cells contributed by the young donor vs the aged host (e.g. essentially all of the naïve T cells derive from the young donor and ~all of the ABCs come from the aged host) but the narrative of this paper was disjointed so it was not clear what the overall intention of the paper was. A good example of this is the final experiment, which shows that BM transplantation of NHD13tg mice with WT BM cells effectively prevents hematological malignancies from developing out to 24 mo of age. But it is not clear how this fits with the narrative of the paper, which is largely around how the aged environment affects young HSC engraftment and differentiation. These mice were engrafted with young BM as young adult mice (2 mo). Doesn't this experiment simply show that you've replaced the defective BM with young healthy WT BM? Also, this experiment answers a different question entirely that centres around the persistence of young BM through the aging process, rather than the engraftment of BM in an aged environment.

- While the study clearly showed a strategy that worked well for engraftment of young cells in aged mice, it did not demonstrate how the efficacy of that strategy compared between aged and young mice (Fig 4.b) This would have been interesting as multiple lines of evidence within the dataset presented suggested that poorer engraftment in aged mice may not arise as a consequence of a hostile aged environment but is secondary to a reduced ability to deplete cells from aged mice. This seems evident from several earlier pieces of data, e.g. Fig 1d (HSC) and Extended data Fig 1c. Comparison of aged and young mouse engraftment after both conditioning regimens might have gotten at where the issue is for aged mice.

- I don't completely agree with the contention that there is nothing inherently obstructive to engraftment in an aged environment (line 175-177 "...young HSCs can successfully engraft in an aged environment, which does not inherently harm HSCs or hinder their ability for proliferation or long-term multilineage reconstitution."). In figure 3C, in contrast to the interpretation, there appears to be a greater proliferation of HSCs in young hosts (many fewer undivided, higher mean division number) compared to aged in the unconditioned mice, and also in higher mean division number in the CD45-SAP conditioned mice. Consolidated data from multiple mice should be shown here, with stats, to support the conclusion of no difference. Thus, while the interpretation of Fig 3e and f may be true, ie. That multilineage reconstitution derived from proliferating young HSCs occurs similarly after transfer to young and aged recipients, the HSC proliferation itself may be the limiting factor in the aged environment.

- Numerous studies have emerged that suggest that inflammation within the aged environment is directly responsible for myeloid skewing of aged BM (PMID: 30799276). A brief discussion on the potential impact of CD45-SAP on inflammation within the aged environment might be good as a potential mechanism for increased engraftment following CD45-SAP conditioning.

- Regarding lines 232-239, the paper being referred to was specifically depleting myeloid biased HSCs (increased in aged mice) to augment the lymphoid defect in aged mice. It is not clear to me how the use of CD45-SAP, which would deplete all leucocytes by targeting a marker that is not increased with age, is a parallel to this experiment. And the statement 'using CD45-SAP for specific HSC depletion' is incorrect as it is not specific for HSCs. Moreover, I cannot see this comparison

(CD45-SAP depletion vs steady state) in Extended Data Fig. 2a) as reported. Are the authors referring to just CD45-SAP treatment (as is indicated in the text) or additionally transplanted with young HSCs? In addition, why doesn't the legend to Extended Data Fig. 2 mention anything about CD45-SAP? This is very confusing.

- This section is misleading "We observed significant reductions in host memory CD4 and CD8 T cells, follicular B cells, and a decrease in ABCs post-treatment. Although there was no increase in naive T cell production, this treatment showed promise in boosting immature B cell generation from aged host cells (Extended Data Fig. 2b-d)." In fact, there was not a significant decrease in ABCs and there were highly significant and substantial decreases in naïve CD8 and CD4 T cells. Which also makes me wonder where this interpretation in the Discussion comes from "we observed a notable increase in thymic seeding and/or expansion, which correlated with dramatically elevated numbers of naive peripheral T cells."

- Concerning the observed discrepancies between the RNA-seq dataset with that from Young et al. 2021. Could the observed difference be simply due to the sampling of different populations? In the Young et al. 2021 paper it appears sequencing was done on LT-HSC ("LT-HSC (SLAM) (Lin- Sca-1+ c-Kit+ Flt3- CD150+ CD48-), LT-HSC (EPCR) (Lin- Sca-1+ c-Kit+ CD34- EPCR+), LT-HSC (CD41+) (Lin- Sca-1+ c-Kit+ Flt3- CD150+ CD48- CD41+)") rather than just the MPP Ly population of cells ("Lineage-cKIT+SCA-1+IL7Ra-CD135+CD150-"). Is it possible that the other populations of cells included in their analysis are driving most of the changes they observe? In this regard is this a fair comparison to draw between datasets?

Technical points:-

- Several of these experimental groups have few mice (as few as 2 in one group) and appear to have been performed only once (E.g. Fig 1b-d, Extended data fig 1). All experiments should have been performed at least twice and the number of times recorded in the figure legends. Moreover, in some instances it is not accurate to report 'no difference' when statistics are applied to groups of 3 where there is either a trend and/or an outlier (e.g. Fig 1b-d, 3b young v aged with young donor).

- It would be useful for Figure 1 to be more thoroughly set up in the text and interpreted. What is the take away from this? How does the effect of CD45-SAP relate to other studies performed in young mice? What's the general summation of this in aged mice?

Reviewer #4

(Remarks to the Author)

Version 1:

Reviewer comments:

Reviewer #1

(Remarks to the Author)

The authors have addressed all of the reviewer's concerns. The addition of key data and clarification of the conclusion within the text has significantly strengthened the manuscript. The reviewer particularly appreciated the addition of the limitations section, which nicely describes the experiments and issues that remain. Overall, this is a strong manuscript that proposes a new regimen that may be able to support more robust and efficient engraftment following BM transplants, particularly in the elderly population. In summary, the reviewer supports the publication of the manuscript.

Reviewer #2

(Remarks to the Author)

The authors have revised the MS in good faith in response to the referees' comments and therefore agree that this paper deserves recognition.

Reviewer #3

(Remarks to the Author)

This issue was highlighted in my initial review and it has not been altered. It is simply not accurate:-

Lines 172-174 "CTV dye dilution revealed that transplanted HSCs exhibited similar proliferation kinetics in both young and aged unconditioned recipients (Fig. 3c and 3d). Even after CD45-SAP treatment, which accelerated HSC proliferation, this similarity persisted across different age environments."

My initial review read as follows:-

"In figure 3C, in contrast to the interpretation, there appears to be a greater proliferation of HSCs in young hosts (many fewer undivided, higher mean division number) compared to aged in the unconditioned mice, and also in higher mean division number in the CD45-SAP conditioned mice. Consolidated data from multiple mice should be shown here, with stats, to support the conclusion of no difference."

Not only has the consolidated data not been supplied, the text stating similarity (above) remains in the manuscript and is not

accurate. The modified statement below does not address these issues. It just neglects to mention that there is any difference. It is misleading to make a comparison, observe differences and then neglect to acknowledge those differences. Moreover, the adoptive transfer only of proliferating cells in Figure 3C likely biases the comparison, as noted previously, since cells in an aged environment do not proliferate as well.

“Together, these results demonstrate that young HSCs can successfully engraft in an aged environment with a preserved capacity for long-term, multilineage reconstitution.” (lines 183–184)

The authors need to reconcile Figs 1 and 3 better. Fig 1 says that the CD45-SAP conditioning is less effective in advanced age, leading to poorer engraftment. Figure 3 tackles the question of whether the poorer engraftment is due to reduced homing or a toxic environment?? My first thought is that it is a problem with the ineffective CD45-SAP conditioning in aged mice as shown in Fig 1. I assume Fig 3 is included to eliminate issues other than with the conditioning, ie. the engraftment.

Again, I refer to my original assessment of the final experiment. The paper is focused on identifying successful strategies to promote engraftment of young HSCs in an aged environment. The final experiment has nothing to do with this. Although the authors try to make it fit by stating that it addresses an age-related hematological disorder, they are correcting a genetic defect in young mice with young bone marrow. This is completely irrelevant to the engraftment of aged HSCs in young mice. The appropriate experiment would have been to wait until the mice aged, and then tried to correct the defect with transplantation.

The authors rebut “we believe it complements the overall narrative by addressing the persistence and functional efficacy of young donor BM over the aging process. This provides valuable insights into the long-term therapeutic benefits of the approach.”

This experiment is not doing this. The young BM ages with the recipient, so when these NHD13tg mice age, their engrafted BM is also old but it is old BM that has a correction for the genetic defect. This does not show that youthful BM is beneficial in an aging context.

original review:-

“...the final experiment, which shows that BM transplantation of NHD13tg mice with WT BM cells effectively prevents hematological malignancies from developing out to 24 mo of age. But it is not clear how this fits with the narrative of the paper, which is largely around how the aged environment affects young HSC engraftment and differentiation. These mice were engrafted with young BM as young adult mice (2 mo). Doesn't this experiment simply show that you've replaced the defective BM with young healthy WT BM?”

Reviewer #4

(Remarks to the Author)

Version 2:

Reviewer comments:

Reviewer #3

(Remarks to the Author)

Apologies for misunderstanding the transfer of undivided rather than divided cells. I still would argue that the final experiment doesn't hold the relevance that the authors claim it does. But they have made a substantial effort to address concerns here.

Reviewer #4

(Remarks to the Author)

Response to Reviewers: Manuscript NCOMMS-24-38839-T

We sincerely thank the Editor and reviewers for their thoughtful and constructive feedback, which we believe has greatly enhanced the quality of our study. We have carefully addressed all suggestions and performed additional experiments and analyses to thoroughly respond to the concerns raised. Below, we provide detailed, point-by-point responses to each of the reviewers' comments.

Reviewer #1 (Remarks to the Author):

Aging is associated with a decline in hematopoietic fitness and increased risk of hematological malignancies, in part arising from defects within hematopoietic stem cells (HSC). These HSC defects manifest clinically as deficient lymphopoiesis and a decrease in adaptive immunity. In the submitted manuscript entitled "A Non-Invasive Stem Cell Therapy Boosts Lymphopoiesis and Averts Age-Related Blood Diseases in Mice" by Konturek-Ciesla et al, the authors develop and optimize a non-myeloablative HSC transplantation protocol wherein HSC from young mice are adoptively transferred into aged mice that are conditioned with a combined CD45 depletion (CD45-SAP) and G-CSF mobilization strategy. Utilizing this protocol, the authors demonstrate successful engraftment of young HSC into aged recipients. They demonstrate an improvement in phenotypic lymphopoiesis parameters by flow cytometry in aged mice. Furthermore, they demonstrate the clinical utility of this protocol by transplanting wild type HSC into a myelodysplasia mouse model (2 month-old NHD13 mice) and tracking the incidence of blood disorders over a course of two years.

A handful of studies in the past decade have demonstrated alternative strategies to achieve successful HSC engraftment without utilizing myeloablative (chemotherapy/irradiation) conditioning regimens. However, the majority of these studies focused on young recipients. It is generally accepted that achieving successful engraftment in aged recipients is significantly more challenging than in younger recipients. Elderly recipients also experience higher toxicity with myeloablative regimens. Only one prior study has demonstrated successful engraftment in aged mice utilizing non-myeloablative regimens. Given the dearth of knowledge in non-myeloablative conditioning in the elderly demographic, the presented study will provide valuable information to scientists working in a wide ranging of disciplines including HSC transplantation, aging, and blood cancers, making this a good candidate to be considered for publication in Nature Communications. The experiments are generally well-designed and well-controlled. However, some statements and claims made by the authors are overstated and not supported by the data presented. The following concerns need to be addressed before being considered for publication.

Response: We thank the reviewer for recognizing the significance of our study to the research community focused on HSC transplantation, aging, and blood cancers. We appreciate the thoughtful feedback, which has prompted us to refine our discussion and perform additional experiments to improve the accuracy and clarity of our data interpretation.

Major Concerns:

1) The primary objective of the manuscript was to develop a safe and efficient conditioning protocol for transplanting young HSCs into an aged recipient. While secondary transplant data was provided for young conditioned recipients (Fig. 2F), it was not provided for the most critical group: aged conditioned recipients (either CD45-SAP alone or CD45-SAP plus G-CSF). In light of the impact of SAP on HSC cell cycle (Fig. 3D), secondary transplant data are crucial to evaluate the long-term safety and efficacy of this conditioning regimen on aged recipients, and their comparison with non-conditioned EE100/500 transplant recipients, steady state controls, and if feasible TBI transplant controls. If these data are not available, please include a Limitations section acknowledging that the impact of SAP procedure on HSC self-renewal in aged recipients remains to be evaluated.

Response: We appreciate the reviewer's suggestion regarding serial transplantation, a method long considered the gold standard for assessing HSC self-renewal *in vivo*. However, the primary aim of our study was to improve hematopoiesis in aged recipients rather than to evaluate HSC activity beyond the lifespan of the transplanted recipients. Furthermore, our data demonstrate that young HSCs transplanted into aged recipients perform comparably to those recovered from young recipients, even though the latter were not conditioned with the combined CD45-SAP/G-CSF+AMD3100 regimen (Fig. 3F).

That said, we recognize the relevance of the reviewer's concern, particularly in the context of prior approaches in the field. To address this, we have included a Limitations section in the manuscript, explicitly acknowledging that the long-term effects of the CD45-SAP procedure on extensive HSC self-renewal in aged recipients remain to be evaluated (lines 393-397).

2) One of the hallmarks of HSC aging is a myeloid-biased output at the expense of lymphopoiesis, both at steady state and following transplantation (Rossi et al PMID: 15967997). In Figure 4B, it appears that G-CSF causes an increase in myeloid-biased engraftment (higher myeloid chimerism as compared to total chimerism). Please include %myeloid cells, %B cells, and %T cells as a frequency of engrafted donor cells as described in PMID: 15967997 for Fig. 1F, 2D, 2H, 3F, and 4B. These reanalyzed data will provide valuable information as to whether the conditioning regimens (CD45-SAP, G-CSF etc) have any impact on myelopoiesis and lymphopoiesis capacity of transplanted HSC.

Response: As suggested by the Reviewer, we have included data in the revised manuscript that displays the lineage distribution within donor (young) peripheral blood cells. This reanalysed data provides valuable insights into the myelopoiesis and lymphopoiesis capacities of transplanted HSCs under the different conditioning regimens (CD45-SAP, G-CSF, etc.). The new data is incorporated into the figures listed below:

- **Supplementary Figure 1d** – related to Fig. 1F
- **Supplementary Figure 2a** – related to Fig. 2D
- **Supplementary Figure 2b** – related to Fig. 2H
- **Supplementary Figure 2c** – related to Fig. 3F
- **Supplementary Figure 3d** – related to Fig. 4B

3) Anemia is a characteristic feature observed in aged C57BL6 mice. Extended data 1A shows that CD45-SAP causes anemia in the short term. Does this resolve over time? And does transplanting young HSC into aged recipients able to improve anemia (Fig. 4A)? Along these lines, complete blood counts in long term engrafted aged SAP/SAP-G-CSF recipients would significantly add to the manuscript.

Response: We thank the reviewer for raising this important point regarding anemia in aged C57BL/6 mice. While anemia has been reported in some studies, reference data from the Jackson Laboratory (<https://www.jax.org/-/media/jaxweb/files/jax-mice-and-services/phenotypic-data/aged-b6-physiological-data-summary.pdf>) do not support anemia as a consistent characteristic of aged C57BL/6 mice. Similarly, prior studies, such as those by Magnani et al. (PMID: 3347096), indicate that anemia is not consistently observed across other mouse strains (e.g., BALB/C).

In our study, we observed reductions in RBC and hemoglobin counts in aged mice, and shortly after CD45-SAP treatment (Supplementary Fig. 1a). In response to the reviewer's comment, we reanalysed blood parameters in aged mice long-term following CD45-SAP treatment and HSC transplantation. Additionally, we have included complete blood count data for long-term engrafted aged SAP/SAP-G-CSF recipients. These data are now presented in Supplementary Fig. 3a-c and discussed in the corresponding Results section (lines 196-199).

4) The number of animals analyzed are very small for some experiments (eg., Fig.1A N=3-4/group; Fig.3A,B 2-3/group). While this is generally not a major concern when the effect sizes are large, it becomes problematic when the observed changes are small. For example, in Fig 3B, it appears that the number of recoverable donor HSC is lower in aged recipients when compared to young recipients in both unconditioned (P=0.11) and conditioned (P=0.34) settings. The authors interpret this data as "When analyzing CD45-SAP-conditioned young and aged hosts, we observed similar amounts of recoverable young donor HSCs (Fig. 3b). This demonstrated that compromised homing/engraftment in aged mice was unlikely to explain the inefficient reconstitution from young HSCs (Fig. 1)." This interpretation is problematic as the N is too small to make the conclusion that aging is associated with no homing defects. Moreover, this is not the ideal experiment to assess homing. The authors should either examine homing at short time points (eg 18 hours) or rephrase this statement to acknowledge these caveats.

Response: We appreciate the reviewer's feedback regarding the small sample sizes in certain experiments (e.g., Fig. 1A and Fig. 3A, B). We agree that small sample sizes can be problematic, particularly when observed changes are subtle. We acknowledge that in Fig. 3B, while the p-values

suggest trends ($P=0.11$ in unconditioned and $P=0.34$ in conditioned settings), the differences between aged and young recipients did not reach statistical significance.

Regarding the comment on assessing homing defects, we agree that studying homing in vivo is challenging, and to the best of our knowledge, there are no universally accepted assays in the field that can provide a definitive consensus. For instance, there are reports that transplanted HSCs can be recovered from the BM of unconditioned recipients, yet long-term reconstitution is very marginal in this setting. This suggests that functional homing is not necessarily reflected by the mere ability to recover HSCs from the BM after transplantation. Our experiment design was based on assessing the functional capacity of transplanted HSCs as a proxy for successful homing and integration rather than relying solely on short-term recovery data.

We acknowledge that our results could, in part, reflect lower engraftment rather than differences in initial homing. However, we believe that our approach remains valid for assessing the functional outcomes of young donor hematopoiesis in aged recipients. That said, we have reworded the manuscript to clarify these points and emphasize that while our findings suggest compromised homing/engraftment is unlikely to be a major factor, the limitations in sample size and experimental design must be considered when interpreting these results (lines 405-406).

5) Fig. 3C-D. CTV experiments clearly show that CD45-SAP conditioning causes a significant increase in HSC proliferation and a complete loss of HSC quiescence (20-30% quiescent in unconditioned vs 0% in CD45-SAP at 2-4 weeks). Were these experiments performed at the same time point for all groups (young, aged, conditioned, unconditioned) or some at 2 weeks and some at 4 weeks? Does the proliferation in SAP groups resolve by 6 weeks (Fig. 3E)? These are important to know because excessive proliferation can have deleterious impact on HSC self-renewal in the long-term.

Response: We conducted our analyses at different time points (2 and 4 weeks after transplantation) across the evaluated groups. To account for division-independent CTV label dilution at these intervals, we co-transplanted animals with CTV-labelled HSCs and CD4⁺ spleen cells, the latter serving as a low-proliferation positive control. This approach allowed us to clearly define the CTV-positive signal in each animal at different time points (an example histogram with CD4⁺ cells is shown in Fig. 3C).

We wish to clarify that the primary recipients in Fig. 3E-F were unconditioned. This was intentional, as at the 6-week time point, we were able to retrieve undivided, CTV-positive HSCs in unconditioned animals, in contrast to CD45-SAP-conditioned primary recipients. Furthermore, we did not present proliferation data for the CD45-SAP groups at 6 weeks post-transplantation because these analyses revealed no detectable CTV labeling, consistent with the irreversible loss of CTV labeling observed at earlier time points (2 and 4 weeks). To improve the clarity of our experimental setup, we have now included a more detailed description of Fig. 3 in the corresponding figure legend (lines 787-792).

We appreciate the reviewer's comment regarding whether HSCs might return to quiescence following initial extensive proliferation. Unfortunately, our current experimental design lacked a suitable tracking system to address this question directly. However, we acknowledge the importance of this issue and have in the revised manuscript, added this concern in our Discussion (lines 362-366).

In light of the reviewer's insight regarding the long-term impact of excessive proliferation on HSC self-renewal, we note that our secondary transplantation experiments (Fig. 2F-H) revealed reduced competitive ability of HSCs initially transplanted into young CD45-SAP-conditioned primary recipients. Combined with the complete loss of CTV labeling in CD45-SAP-conditioned hosts (Fig. 3C-D), these findings suggest that the extensive proliferation of donor HSCs associated with CD45-SAP conditioning may impair their reconstitution potential in secondary recipients.

6) In Fig.3F, please include total donor chimerism as reported for other transplant data.

We appreciate the reviewer's suggestion to include total donor chimerism in Fig. 3F. However, we chose not to provide this in the main figure because our secondary transplantation experiment yielded some negative (non-myeloid-reconstituted) animals. As such, we consider this experiment to function more as a limiting dilution assay, and we believe the current representation - reporting the frequency of multilineage or lymphoid-reconstituted mice - provides a clearer overview of the overall HSC behavior.

That said, to address the reviewer's comment, we have included total donor chimerism and lineage distribution within donor cells as supplementary information. This additional data is presented in **Supplementary Fig. 2c**, alongside related analyses addressing Reviewer's Comment 2.

7) For data in Fig. 4E-H and Extended Data 2, data are presented as 'normalized to young' which makes it difficult to evaluate the extent of lymphopoiesis restoration. Non-normalized absolute values for these parameters in steady state controls (young and aged unmanipulated controls), and conditioned aged recipients would be necessary to illustrate two points: age-related defects in lymphopoiesis and degree of reversal following stem cell therapy. These non-normalized data as bar graphs can be added in extended data.

We thank the reviewer for this valuable suggestion, which aligns closely with the main conclusions of our study. In response, we re-evaluated our previous analyses, where data were normalized to young mice purchased from a commercial vendor. Upon closer inspection, we observed that the aged mice in these cohorts did not exhibit the full extent of "aged" phenotypes described in prior literature (e.g., PMID: 29891535, PMID: 31447053). Specifically, while reductions in levels of naive lymphocyte were evident, these were not as pronounced as reported in other studies. This discrepancy may be partly due to the relatively younger age of the reference mice in our prior analysis (18–19 months) compared to the animals used in our current experimental models, as well as differences in the origin of the mice.

To address this concern and enhance the rigor of our study, we analyzed young (n = 6) and aged (n = 7) cohorts derived from our own animal facilities. These new analyses support our original conclusions: aged mice exhibit a reduced output of naive lymphocytes and an increase in age-associated B cells (ABCs), while transplanted young HSCs preferentially contribute to immature and naive lymphoid subsets. While the absolute numbers of these lymphocytes remain lower than those in young mice - likely due to thymic atrophy in aged hosts and the retention of host ABCs - we believe that the ability of transplanted cells to supply immature and naive lymphocytes could substantially support the aged immune system by providing cell types that it otherwise generates poorly at advanced ages. This new data has been included in our revised manuscript as **Supplementary Fig. 3e-h**.

We agree that a direct investigation of immune responses in aged mice transplanted with young HSCs, including quantification of responses from host versus donor lymphocytes, would further strengthen these conclusions. However, given the scope of the current study, we have opted to leave this detailed immune response analysis for future investigations. To acknowledge this limitation, we have included a discussion of this point in the "Limitations" section of the revised manuscript (lines 398-401).

8) The authors make the statement that "Taken together, these results establish that young HSCs can successfully engraft in an aged environment, which does not inherently harm HSCs or hinder their ability for proliferation or long-term multilineage reconstitution." While the authors clearly show that young HSCs can successfully engraft an aged environment, they do not have adequate data to support that the aged environment does not harm or hinder the transplanted young donor HSCs. Data in Fig. 3D contradicts this statement (% quiescent HSC in young vs aged unconditioned recipients). This difference in quiescence (young vs aged unconditioned), and the impact of SAP conditioning on HSC proliferation needs to be discussed in the manuscript.

Response: We thank the reviewer for raising this important point and appreciate the opportunity to clarify our conclusions.

We agree that donor HSCs transplanted into young and aged mice exhibit differences in quiescence, with HSCs in aged recipients showing a higher degree of quiescence compared to those in young hosts (Fig. 3D). While our data demonstrate that HSCs recovered from both young and aged recipients support long-term hematopoiesis, additional analyses (Reviewer #1, Comment 6) revealed that the magnitude of multilineage reconstitution was, on average, higher for cells recovered from aged primary hosts compared to young hosts (average $22.9\% \pm 14.0\%$ for aged vs. $3.9\% \pm 3.2\%$ for young primary recipients, Supplementary Fig. 2c). This finding suggests that the quiescent state of HSCs in aged recipients may help preserve their functionality, enabling robust activity when exposed to other environments, such as young, irradiated hosts.

We emphasize that our key conclusion is that the aged environment per se does not impair the capacity of transplanted young HSCs to support multilineage blood generation over the long term. However, we recognize that the observed differences in quiescence and the potential impact of CD45-SAP conditioning on HSC proliferation warrant further discussion. In response to the reviewer's comment, we have expanded on these points in the revised manuscript (lines 355-366) and modified the sentence in question to:

“Together, these results demonstrate that young HSCs can successfully engraft in an aged environment with a preserved capacity for long-term, multilineage reconstitution.” (lines 183-184)

9) In their RNA Seq analyses, the authors conclude that “Together, these results revealed no significant differences in the transcriptomic signatures of donor MPP Ly cells, even when exposed to an aging environment.” However, the comparisons performed and controls included are insufficient to support this statement. If the conditioning regimen caused a large scale shift in the transcriptome, it could also explain the observed lack of significant gene expression changes across the 4 groups (all of which received conditioning). It would be worthwhile to perform comparisons of the 1) donor MPP (young) vs host MPP (aged), and 2) donor MPP (young) vs host MPP (young) which might provide some information as to whether the aging environment impacts MPP gene expression. Ideally, MPP from steady state (non-conditioned, non-transplanted) young and aged mice would need to be compared alongside. However, if these steady state data are not available, the interpretations should be performed in light of these caveats. Additionally, please include the time point after the transplant that the RNA Seq was performed and its experimental set-up in Extended Data Table 3.

Response: In our revised manuscript, we have included the suggested pairwise comparisons between donor- and host-derived MPP Ly cells. While these additional analyses identified a few more differentially expressed genes compared to our previous comparisons, they did not reveal any molecular pathways indicative of mechanisms driving multipotent progenitor cell aging. This could suggest either that aging has minimal impact on these cells or that the treatment restored their molecular characteristics.

However, we acknowledge that the absence of untreated, steady-state controls limits the strength of our conclusions. This limitation has been addressed in the Limitations section (lines 402–405). Additionally, we have clarified the time point after transplantation when the RNA-seq was performed and included the experimental setup in **Supplementary Figure 4** for greater transparency.

Minor concerns:

1) The title is overly broad. It implies that the proposed stem cell therapy averts all aging related blood diseases in mice. It needs to be made specific to the data presented in the manuscript. HSC transplantation by its nature is an invasive procedure. Consider changing non-invasive to “non-chemo/irradiation based” based or “non-myeloablative” or “low-intensity conditioning”.

Response: We appreciate the reviewer’s suggestion regarding the title and agree that it should be more specific to the data presented in the manuscript. Regarding the use of the term “non-invasive,” we recognize that “non-myeloablative” could cause confusion, as it is associated with established clinical regimens that still involve some chemotherapy treatments. To avoid this potential confusion while accurately conveying the reduced intensity of our approach compared to traditional conditioning regimens, we propose using the term “non-genotoxic” instead. We believe this term better reflects the nature of the procedure, which avoids classical chemotherapy or irradiation and is less harsh on the recipient while still achieving effective conditioning.

2) The authors provide no functional readouts for improvements in lymphopoiesis (e.g vaccine challenge etc) or HSC self-renewal in aged conditioned recipients. Consequently, please tone down some of the statements including “not only survive but thrive in aged hosts, dramatically improving hematopoietic output and ameliorating age-compromised lymphopoiesis”.

Response: We appreciate the reviewer’s comment and have revised the statement in the abstract to more accurately reflect the data presented in the manuscript (lines 15–16). This updated phrasing ensures that the claims align with the scope and findings of the study.

3) In the Discussion the authors state that “If such age-related alterations are significant, introducing young HSCs into aged recipients might not effectively restore youthful hematopoietic function...Firstly, we demonstrate that young HSCs transplanted into aged hosts exhibit behavior akin to their performance in a normal environment.” To justify this statement, a comprehensive hematopoietic analysis needs to be performed (HSC self-renewal assays, comprehensive blood counts, HSC RNA Seq, functional assays for evaluating immunosenescence, evaluation of BM niches etc). If these data are not available, this statement should be toned down a bit.

Response: We thank the reviewer for this insightful comment and agree that it is important to avoid overstating the data. In response, we have revised the statement in lines 352–354 to better reflect our findings, as follows:

“Our findings further support this by demonstrating that young HSCs transplanted into aged hosts behave differently from aged host HSCs, including displaying a more robust lymphoid output.”

4) A prior study demonstrated successful engraftment of young HSCs in aged mice (Reference 27, Guderyon, M.J., et al). In the Discussion, it would be useful to compare and contrast the CD45 SAP/GCSF protocol with this published study. Additionally, the relative advantages and disadvantages of CD45-SAP compared with other strategies cited in their references would add value in placing this protocol in the broader context.

Response: We thank the reviewer for this suggestion. In the revised manuscript, we have highlighted the similarities and differences between the CD45-SAP/G-CSF protocol and the study by Guderyon et al. (Reference 27) in the Discussion (lines 305–311).

5) The authors state that “We and others have previously established the efficacy of a polyvinyl alcohol (PVA)-based culture system in promoting murine HSC expansion”. Please rephrase this to accurately credit Wilkinson et al for the discovery of PVA-based expansion. There should be no problem in having a second sentence stating that “we have validated this expansion platform in our prior study”.

Response: We thank the reviewer for this comment and agree on the importance of clearly recognizing prior contributions. We have revised the sentence to accurately attribute Wilkinson et al. for the discovery of the PVA-based culture system. The revised text now reads:

“Previous work by Wilkinson et al. established the efficacy of a polyvinyl alcohol (PVA)-based culture system in promoting murine HSC expansion¹¹, which we have validated in our prior work¹².” (lines 102-104).

6) The authors state that “Conversely, the reconstitution levels were markedly decreased upon competitive transplantation, mirroring the reduction in HSCs derived from primary TBI-conditioned hosts (Fig. 2h)”. It is not clear as to which reduction in HSCs are being referred to here.

Response: We thank the reviewer for pointing out this ambiguity. To improve clarity, we have revised the statement to explicitly refer to the reduction in HSC activity derived from primary TBI-conditioned hosts. The updated text now reads:

“Conversely, the reconstitution levels were markedly decreased upon competitive transplantation, mirroring the reduction in HSC activity derived from primary TBI-conditioned hosts (Fig. 2h).” (lines 140-142)

7) Please include flow cytometry gating strategy for all flow cytometric analysis in Supplemental Data.

Response: We have included the gating strategies for all flow cytometric analyses in Supplementary Figures 5 and 6.

Reviewer 2

The authors proposed a mouse model in which the CD45-SAP condition is used to treat hematologic diseases caused by aging HSCs by transplantation of young HSCs. The authors further developed their experimental results to ultimately treat MDS. On the other hand, there is great concern about the authors' new finding. The authors have a major concern about the new regulation, which, as they note in their interim summary, is that it does not offer anything new in comparison with previous reports. As I understand it, they then go on to manipulate the transplant system in a complex way from Fig. 3, but I see no particularly surprising findings. In addition, some concerns are noted below.

We appreciate the reviewer's feedback and would like to address the comment regarding the perceived lack of novelty and the mention of "new regulation." It appears there may be some misunderstanding, as our study does not pertain to any new regulatory mechanisms. Instead, we present a novel therapeutic model that utilizes CD45-SAP to specifically target and deplete aged HSPCs, offering a clear advancement over existing transplantation methodologies.

While prior studies have explored HSC transplantation, our approach introduces a targeted and efficient conditioning strategy that enhances the engraftment of young donor HSCs in aged or diseased recipients. This innovation addresses a critical gap in aging HSC biology, particularly in rejuvenating hematopoiesis in older individuals. Importantly, our findings extend beyond transplantation outcomes, providing a potential therapeutic avenue for age-related hematologic conditions, such as MDS, that has not been demonstrated in earlier reports.

We hope this clarification resolves any misunderstanding and emphasizes the novelty and clinical relevance of our study.

Major Concerns:

1) That's interesting that Old-HSC are not reduced by CD45-SAP compared to Young-HSC, but why are Old-HSC not reduced more than Young-HSC? Is it because of optimized conditions?

Response: We thank the reviewer for this insightful comment. Our study demonstrates that CD45-SAP treatment significantly reduces HSC numbers in both young and aged animals. The observed differences in residual stem cell numbers post-treatment, with aged mice retaining more HSCs, are likely attributable to their initially higher absolute HSC counts compared to young mice.

Additionally, other mechanisms may contribute to the incomplete depletion of host HSCs in aged subjects. For example, an increased uptake of CD45-SAP by peripheral cells in aged mice could limit the availability of active immunotoxin in the bone marrow. Alternatively, CD45-SAP might selectively target specific subsets of aged HSCs, leaving certain populations less affected. This possibility aligns with Reviewer 1's comments on the functional properties of aged HSCs following treatment.

However, as the primary focus of our study was to improve overall hematopoiesis in aged subjects rather than to dissect specific HSC subsets, these observations underscore the need for further investigations. In response to the reviewer's comment, we have included a discussion on the potential mechanisms underlying reduced HSC depletion in aged mice in the revised manuscript (lines 336–344).

2) CD45-SAP as a model for aging hematopoiesis or MDS treatment. Auto's system seems inappropriate. What would happen in the experimental system of Allo's system (at least haplo-MHC mismatch)?

Response: We appreciate the reviewer's suggestion to consider an Allo-transplant model (e.g., involving a haplo-MHC mismatch) for investigating the effects of CD45-SAP on aging hematopoiesis and MDS treatment. However, we intentionally focused on a syngeneic transplant model in this study for several reasons.

Our primary objective was to evaluate the effects of the CD45-SAP conditioning regimen in a controlled setting, minimizing immunological variables to isolate its impact on hematopoietic function. Introducing an Allo-system, particularly with a haplo-MHC mismatch, would have added complexities such as graft-versus-host disease (GVHD) and other immune-mediated effects, which are beyond the scope of this study. While an Allo-model could provide valuable insights into immunological interactions, it would not align with our goal of understanding the direct effects of CD45-SAP in the context of aging hematopoiesis and syngeneic transplantation.

We agree that exploring Allo-transplant models in future studies could further extend the applicability of this approach, particularly in the context of MDS treatment.

3) Related to 2), are CD45-SAP transplants rejected by donor cells in allo-HSC transplants?

Response: We thank the reviewer for this comment. As mentioned in our response above, our study focuses on syngeneic transplants to minimize immunological variables and isolate the effects of the CD45-SAP conditioning regimen. However, prior work from other groups has addressed the potential for rejection in Allo-HSC transplant settings. We refer the reviewer to the following publications for additional insights: PMID: 34730109 and PMID: 38447038.

4) Finally, the novelty (point) of the authors did not come across to me, as CD45-SAP is a paper that is quite dependent on and very similar to the previously reported paper by Palchaudhuri, R et al. If there is really a major novelty, it might be good to prepare a summary with figures that we see more often these days.

Response: We appreciate the opportunity to highlight the novel aspects of our study in comparison to the work by Palchaudhuri et al. While their study provided valuable insights into CD45-SAP-mediated HSC depletion, it focused on younger hosts and did not address the distinct challenges posed by aged recipients. In contrast, our work specifically targets aging HSCs, which are associated with well-documented functional decline and an increased risk of hematologic disorders, such as myelodysplastic syndrome (MDS).

The novelty of our study lies in its demonstration that CD45-SAP can condition aged hosts to successfully accept young HSC transplants, enabling significant hematopoietic recovery in a biologically distinct and more challenging environment. This represents a critical advancement, as no evidence from Palchaudhuri et al. suggests that their method is effective in aged hosts, where the biological context differs substantially from that of younger subjects.

Additionally, our focus on MDS as an age-related disease with limited treatment options further differentiates our work. By tailoring the CD45-SAP approach to aged populations and demonstrating its potential therapeutic benefit in this clinically relevant setting, our study extends beyond the methodology presented by Palchaudhuri et al. and establishes a novel application with clear translational relevance.

Minor Comment:

It is confusing for the reader to tell from the figure which mouse is the donor and which is the recipient.

Response: While this concern was not raised by other reviewers, we recognize the potential for confusion regarding the distinction between donor and recipient mice in our figures. To address this, we have revised the figure legends and added clearer annotations to explicitly indicate donor and recipient mice. These changes aim to improve clarity and ensure that all readers can easily interpret the data. We believe these updates enhance the readability of the figures without affecting the content or conclusions.

Reviewer 3

Major Concerns:

1) This study primarily sought to investigate non-invasive BM conditioning as a regimen to promote the effective engraftment of young HSCs in aged recipients. It ultimately showed that the best results for aged mice were obtained when CD45-SAP was combined with a mobilization strategy to disrupt the niches. It also investigated whether a multiple smaller transplantations of HSCs was superior to a single large HSC transplantation in young mice and finally showed that BM transplantation could reverse an inherent genetic predisposition to hematological malignancies in a transgenic mouse model. The data contain interesting findings. In particular, the analysis of how young BM reconstitutes in a young host vs an aged host, and also the qualitative analysis of cells contributed by the young donor vs the aged host (e.g. essentially all of the naïve T cells derive from the young donor and ~all of the ABCs come from the aged host) but the narrative of this paper was disjointed so it was not clear what the overall intention of the paper was. A good example of this is the final experiment, which shows that BM transplantation of NHD13tg mice with WT BM cells effectively prevents hematological malignancies from developing out to 24 mo of age. But it is not clear how this fits with the narrative of the paper, which is largely around how the aged environment affects young HSC engraftment and differentiation. These mice were engrafted with young BM as young adult mice (2 mo). Doesn't this experiment simply show that you've replaced the defective BM with young healthy WT BM? Also, this experiment answers a different question entirely that centres around the persistence of young BM through the aging process, rather than the engraftment of BM in an aged environment.

Response: We thank the reviewer for recognizing the interesting findings in our study and for the constructive feedback regarding the narrative and scope of the manuscript. We appreciate the opportunity to clarify the rationale and interpretation of the final experiment involving NHD13tg mice.

In response to the reviewer's comment, we have revised the manuscript to explicitly address how this experiment fits within the broader context of our study. Specifically, we clarified that while the primary focus of the study is on the effects of the aged environment on young HSC engraftment and differentiation, the NHD13tg mouse model provides an additional perspective by demonstrating the therapeutic potential of our approach to correct genetic predispositions to hematological malignancies. This highlights the broader relevance of the CD45-SAP-based conditioning strategy, not only for overcoming challenges in aged environments but also for treating genetic disorders associated with hematopoietic dysfunction. These points are now discussed in lines 276-279 and lines 375-379 of the revised manuscript.

While we acknowledge that this particular experiment centres on engraftment in young adult mice rather than aged recipients, we believe it complements the overall narrative by addressing the persistence and functional efficacy of young donor BM over the aging process. This provides valuable insights into the long-term therapeutic benefits of the approach.

We hope these revisions clarify the purpose and scope of the NHD13tg experiments and their relevance to the overarching goals of the study.

2) While the study clearly showed a strategy that worked well for engraftment of young cells in aged mice, it did not demonstrate how the efficacy of that strategy compared between aged and young mice (Fig 4.b) This would have been interesting as multiple lines of evidence within the dataset presented suggested that poorer engraftment in aged mice may not arise as a consequence of a hostile aged environment but is secondary to a reduced ability to deplete cells from aged mice. This seems evident from several earlier pieces of data, e.g. Fig 1d (HSC) and Extended data Fig 1c. Comparison of aged and young mouse engraftment after both conditioning regimens might have gotten at where the issue is for aged mice.

Response: We thank the reviewer for this thoughtful comment. Our data indeed suggest that the reduced efficacy in depleting host cells is a key contributor to the transplantation barrier observed in aged mice. This conclusion is supported by several findings in our study, including Fig. 1d (HSC counts) and Supplementary Fig. 1c, which show incomplete depletion of host cells in aged recipients.

In our experiments, young mice were not subjected to both conditioning regimens because CD45-SAP treatment alone proved sufficient for achieving robust, long-term donor cell engraftment (as shown in Fig. 1d and Fig. 2d). In contrast, aged mice required the enhanced combination regimen with a mobilization-based treatment to achieve comparable engraftment. We acknowledge that a direct comparison of engraftment outcomes in young versus aged mice under both conditioning regimens could provide further insights into the mechanisms underlying the observed differences.

To address the reviewer's point, we have expanded the discussion in the manuscript to explore potential factors that may explain the reduced depletion efficacy in aged mice. Specifically, we discuss the potential role of pharmacokinetics, the immunotoxin's mode of action, and increased cellularity in aged hosts, all of which may influence the availability and effectiveness of CD45-SAP in depleting host cells. These points are now included in the revised discussion (lines 336–344).

We appreciate the reviewer's suggestion and agree that further studies directly comparing young and aged mice under both regimens would be valuable to pinpoint the specific barriers to engraftment in aged recipients.

3) I don't completely agree with the contention that there is nothing inherently obstructive to engraftment in an aged environment (line 175-177 "...young HSCs can successfully engraft in an aged environment, which does not inherently harm HSCs or hinder their ability for proliferation or long-term multilineage reconstitution."). In figure 3C, in contrast to the interpretation, there appears to be a greater proliferation of HSCs in young hosts (many fewer undivided, higher mean division number) compared to aged in the unconditioned mice, and also in higher mean division number in the CD45-SAP conditioned mice. Consolidated data from multiple mice should be shown here, with stats, to support the conclusion of no difference. Thus, while the interpretation of Fig 3e and f may be true, ie. That multilineage reconstitution derived from proliferating young HSCs occurs similarly after transfer to young and aged recipients, the HSC proliferation itself may be the limiting factor in the aged environment.

Response: We thank the reviewer for this comment. As a similar concern was raised by Reviewer 1 (Comment 8), we have revised the manuscript to address the differences in HSC proliferation dynamics observed between young and aged recipients, as well as the impact of host conditioning on donor cell division dynamics (lines 355–366).

To better align the conclusions with the data, we also modified the statement referenced by the reviewer to reflect these nuances:

"Together, these results demonstrate that young HSCs can successfully engraft in an aged environment with a preserved capacity for long-term, multilineage reconstitution." (lines 183–184)

4) Numerous studies have emerged that suggest that inflammation within the aged environment is directly responsible for myeloid skewing of aged BM (PMID: 30799276). A brief discussion on the potential impact of CD45-SAP on inflammation within the aged environment might be good as a potential mechanism for increased engraftment following CD45-SAP conditioning.

Response: We thank the reviewer for this suggestion. In response, we have included a brief discussion in the revised manuscript on the potential impact of CD45-SAP on inflammation within the aged environment and how this might relate to its efficacy in enhancing engraftment compared to more traditional conditioning approaches (lines 355-366).

While it is possible that CD45-SAP could target inflammatory components within the aged environment, such as memory T cells or age-associated B cells (ABCs), our data do not show strong evidence for their depletion. Therefore, while we acknowledge this as an intriguing possibility, further studies would be required to evaluate whether CD45-SAP contributes to engraftment efficacy through modulation of the inflammatory niche.

5) Regarding lines 232-239, the paper being referred to was specifically depleting myeloid biased HSCs (increased in aged mice) to augment the lymphoid defect in aged mice. It is not clear to me how the use of CD45-SAP, which would deplete all leucocytes by targeting a marker that is not increased with age, is a parallel to this experiment. And the statement 'using CD45-SAP for specific HSC depletion' is incorrect as it is not specific for HSCs. Moreover, I cannot see this comparison (CD45-SAP depletion vs steady state) in Extended Data Fig. 2a) as reported. Are the authors referring to just CD45-SAP treatment (as is indicated in the text) or additionally transplanted with young HSCs? In addition, why doesn't the legend to Extended Data Fig. 2 mention anything about CD45-SAP? This is very confusing.

Response: We thank the reviewer for highlighting these important points, and we appreciate the opportunity to clarify and improve the accuracy and clarity of the manuscript and figures.

First, we acknowledge that CD45-SAP is not specific for HSCs but rather targets a broader range of leukocytes. We have rephrased the referenced statement as follows: "Using CD45-SAP to deplete hematopoietic cells..." (line 246).

Second, we have revised the title and legend of Supplementary Fig. 3 (previously referred to as Extended Data Fig. 2) to explicitly indicate the conditions included in the experiment and to specify when CD45-SAP treatment was applied, whether in isolation or in combination with transplantation of young HSCs. This ensures that the figure aligns with the text and resolves any confusion about the experimental design.

Lastly, while the paper referenced in lines 244-246 focused on selectively depleting myeloid-biased HSCs to augment lymphopoiesis, our use of CD45-SAP is broader and not limited to myeloid-biased HSCs. We have clarified that the comparison between the two approaches lies in the shared goal of improving lymphopoiesis, albeit through different mechanisms, and not in the specificity of the depletion strategy.

We hope these revisions address the reviewer's concerns and enhance the clarity and precision of the manuscript.

6) This section is misleading "We observed significant reductions in host memory CD4 and CD8 T cells, follicular B cells, and a decrease in ABCs post-treatment. Although there was no increase in naive T cell production, this treatment showed promise in boosting immature B cell generation from aged host cells (Extended Data Fig. 2b-d)." In fact, there was not a significant decrease in ABCs and there were highly significant and substantial decreases in naive CD8 and CD4 T cells. Which also makes me wonder where this interpretation in the Discussion comes from "we observed a notable increase in thymic seeding and/or expansion, which correlated with dramatically elevated numbers of naive peripheral T cells.

Response: We thank the reviewer for this important comment. As noted earlier in our response to Reviewer 1, Comment 7, our detailed examination revealed that the aged mice used in the original manuscript did not fully exhibit the aging phenotype typically described in the literature (see PMID: 29891535, PMID: 31447053). To address this concern and improve the clarity and scientific rigor of our study, we analyzed new cohorts of steady-state young and aged mice, which aligned more closely with our prior observations.

In the revised manuscript, we have updated the text to reflect these findings more accurately and modified the figure title to enhance clarity. The referenced statements were also revised as follows:

"While the absolute numbers of these cell populations remained lower than in young mice, we still observed reductions in host memory CD4 and CD8 T cells and follicular B cells compared to aged steady-state controls (Supplementary Fig. 3e-h)." (lines 248-250).

"...we observed a notable increase in thymic seeding and/or expansion from the transplanted young grafts, which correlated with elevated numbers of naive donor-derived peripheral T cells." (lines 371 - 373)

Additionally, we have attempted to adjust the discussion to align with these revised findings and support the study's overall conclusions.

7) Concerning the observed discrepancies between the RNA-seq dataset with that from Young et al. 2021. Could the observed difference be simply due to the sampling of different populations? In the Young et al. 2021 paper it appears sequencing was done on LT-HSC ("LT-HSC (SLAM) (Lin- Sca-1+ c-Kit+ Flt3- CD150+ CD48-), LT-HSC (EPCR) (Lin- Sca-1+ c-Kit+ CD34- EPCR+), LT-HSC (CD41+) (Lin- Sca-1+ c-Kit+ Flt3- CD150+ CD48- CD41+)" rather than just the MPP Ly population of cells ("Lineage-cKIT+SCA-1+IL7Ra-CD135+CD150-"). Is it possible that the other populations of cells included in their analysis are driving most of the changes they observe? In this regard is this a fair comparison to draw between datasets?

Response: We thank the reviewer for raising this important point. We acknowledge that the transcriptome analysis by Young et al. was conducted on LT-HSC populations, which differ from the MPP Ly populations examined in our study. As such, we agree that comparisons between these datasets should be interpreted with caution, as the inclusion of different cell populations in their analysis may indeed drive many of the transcriptomic changes observed.

To clarify, we used the Young et al. dataset for two main purposes: (1) to validate our analytical approach and (2) to highlight distinctions between HSCs and downstream progenitors, such as MPP Ly, in terms of age-related transcriptomic changes. However, to avoid confusion and ensure the focus

remains on the populations we studied, we have removed this comparison from the Supplementary Figure. Instead, we incorporated additional analyses as detailed in our response to Reviewer 1, Comment 9, which are now included in Supplementary Fig. 4.

Additionally, we acknowledge the limitation that our study did not include transcriptomic comparisons with non-transplanted, steady-state young and aged mice. This limitation has been explicitly discussed in the revised text (lines 402-405) to ensure transparency in our interpretations and conclusions.

Technical Points:

1) Several of these experimental groups have few mice (as few as 2 in one group) and appear to have been performed only once (E.g. Fig 1b-d, Extended data fig 1). All experiments should have been performed at least twice and the number of times recorded in the figure legends. Moreover, in some instances it is not accurate to report 'no difference' when statistics are applied to groups of 3 where there is either a trend and/or an outlier (e.g. Fig 1b-d, 3b young v aged with young donor).

Response: We acknowledge the small sample sizes in certain experiments, which were partly due to the limited availability of aged animals and logistical constraints. To ensure transparency, we have clearly noted the sample sizes for all experimental groups in the figure legends. Furthermore, we have discussed the limitations posed by these small sample sizes in the revised manuscript (lines 405-406).

Additionally, we recognize the importance of avoiding strong conclusions where data may be underpowered. In such cases, we have refrained from making definitive statements and have instead highlighted trends or the need for further validation. Where applicable, we have revised the text to ensure our interpretations are appropriately cautious and reflect the limitations of the dataset.

2) It would be useful for Figure 1 to be more thoroughly set up in the text and interpreted. What is the take away from this? How does the effect of CD45-SAP relate to other studies performed in young mice? What's the general summation of this in aged mice?

Response: We appreciate the reviewer's suggestion. To increase the clarity of the text and rational, we included a short introduction in lines 64-67 and refined the summary statement in lines 98-99.

REVIEWER COMMENTS

Reviewer #1 (Remarks to the Author):

The authors have addressed all of the reviewer's concerns. The addition of key data and clarification of the conclusion within the text has significantly strengthened the manuscript. The reviewer particularly appreciated the addition of the limitations section, which nicely describes the experiments and issues that remain. Overall, this is a strong manuscript that proposes a new regimen that may be able to support more robust and efficient engraftment following BM transplants, particularly in the elderly population. In summary, the reviewer supports the publication of the manuscript.

We thank the reviewer for taking the time to review our manuscript and for providing constructive feedback that has greatly enhanced our work.

Reviewer #2 (Remarks to the Author):

The authors have revised the MS in good faith in response to the referees' comments and therefore agree that this paper deserves recognition.

We thank the reviewer for evaluating our manuscript and the valuable insights that have improved our work.

Reviewer #3 (Remarks to the Author):

This issue was highlighted in my initial review and it has not been altered. It is simply not accurate:-

Lines 172-174 “CTV dye dilution revealed that transplanted HSCs exhibited similar proliferation kinetics in both young and aged unconditioned recipients (Fig. 3c and 3d). Even after CD45-SAP treatment, which accelerated HSC proliferation, this similarity persisted across different age environments.”

My initial review read as follows:-

“In figure 3C, in contrast to the interpretation, there appears to be a greater proliferation of HSCs in young hosts (many fewer undivided, higher mean division number) compared to aged in the unconditioned mice, and also in higher mean division number in the CD45-SAP conditioned mice. Consolidated data from multiple mice should be shown here, with stats, to support the conclusion of no difference.”

Not only has the consolidated data not been supplied, the text stating similarity (above) remains in the manuscript and is not accurate. The modified statement below does not address these issues. It just neglects to mention that there is any difference. It is misleading to make a comparison, observe differences and then neglect to acknowledge those differences. Moreover, the adoptive transfer only of proliferating cells in Figure 3C likely biases the comparison, as noted previously, since cells in an aged environment do not proliferate as well.

“Together, these results demonstrate that young HSCs can successfully engraft in an aged environment with a preserved capacity for long-term, multilineage reconstitution.” (lines 183–184)

We thank the reviewer for these insightful comments and would like to emphasize that the values for undivided and divided HSCs are provided in Fig 3D and a Source Data file. Our statement was based on these graphs rather than the example plots provided in Fig 3C. Our interpretation here is that we cannot quite share the view that there are “many” more undivided cells, although stats do suggest some significance for undivided cells in the unconditioned

setting. That said, the majority of cells have also divided in setting and so do not support that the aged environments would drive a general energy of introduced HSCs. To address the reviewer's comment, we have revised the text to include the average \pm SD values for each age group and added the grid to the histogram plots in Figure 3C to improve data readability.

In response to the reviewer's comment regarding the adoptive transfer, we would like to clarify that, in our experimental setup, we transplanted undivided HSCs, not "only proliferating cells," as suggested by the reviewer. The decision to use undivided HSCs as donors was based on prior data, which showed that their divided counterparts failed to reconstitute secondary hosts (Figure 1 attached to this response letter).

Figure 1. Undivided HSCs Contribute to Long-Term Multilineage Output. HSCs (Lin-Sca1+cKit+CD48-CD150+CD201high) were expanded ex vivo in a PVA-based culture for 21 days, labeled with Cell Trace Violet (CTV) dye, and transplanted into unconditioned recipients. After 6 weeks, CTV+ and CTV- HSCs were isolated from primary hosts and transplanted into secondary lethally irradiated recipients (10 HSCs + 500K competitor BM per mouse). The dot graph shows the frequency of donor-derived cells in the indicated lineages 16 weeks after secondary transplantation.

The authors need to reconcile Figs 1 and 3 better. Fig 1 says that the CD45-SAP conditioning is less effective in advanced age, leading to poorer engraftment. Figure 3 tackles the question of whether the poorer engraftment is due to reduced homing or a toxic environment?? My first thought is that it is a problem with the ineffective CD45-SAP conditioning in aged mice as shown in Fig 1. I assume Fig 3 is included to eliminate issues other than with the conditioning, ie. the engraftment.

We thank the reviewer for the comment. To enhance the clarity of our aims and the rationale for experiments presented in Figures 1 and 3, we have revised the manuscript text connected to these experiments.

Again, I refer to my original assessment of the final experiment. The paper is focused on identifying successful strategies to promote engraftment of young HSCs in an aged environment. The final experiment has nothing to do with this. Although the authors try to make it fit by stating that it addresses an age-related hematological disorder, they are correcting a genetic defect in young mice with young bone marrow. This is completely irrelevant to the engraftment of aged HSCs in young mice. The appropriate experiment would have been to wait until the mice aged, and then tried to correct the defect with transplantation.

The authors rebut "we believe it complements the overall narrative by addressing the persistence and functional efficacy of young donor BM over the aging process. This provides valuable insights into the long-term therapeutic benefits of the approach."

This experiment is not doing this. The young BM ages with the recipient, so when these NHD13tg mice age, their engrafted BM is also old but it is old BM that has a correction for the genetic defect. This does not show that youthful BM is beneficial in an aging context.

original review:-

“...the final experiment, which shows that BM transplantation of NHD13tg mice with WT BM cells effectively prevents hematological malignancies from developing out to 24 mo of age. But it is not clear how this fits with the narrative of the paper, which is largely around how the aged environment affects young HSC engraftment and differentiation. These mice were engrafted with young BM as young adult mice (2 mo). Doesn't this experiment simply show that you've replaced the defective BM with young healthy WT BM?”

While we understand the concern that this experiment may not directly address the engraftment of aged HSCs in young or aged recipients, we respectfully argue that this final experiment remains relevant to the overall narrative of the study for the following reasons:

1. The NHD13 model reflects certain aspects of aging and MDS. The NHD13 mouse model is widely recognized as a model for myelodysplastic syndromes (MDS), which is primarily a disease of the aged population. Although the mice in this experiment were transplanted at 2 months of age, the genetic defect driving the disease mirrors key features of premature aging and age-associated hematological disorders, including genomic instability, impaired hematopoiesis, and clonal evolution. Thus, while the mice themselves are young at the time of treatment, the model captures segmental aspects of aging-associated disease, making the experiment highly relevant in this context.

2. Prophylactic therapy for age-associated disease. The objective of this experiment was not to correct ongoing disease but to assess the potential of prophylactic therapy in mitigating the development of age-associated hematological malignancies. By transplanting young wild-type (WT) bone marrow (BM) into NHD13 mice, we demonstrate that early intervention can prevent the onset of MDS-like pathology, even as both the recipient and the transplanted cells age. This provides critical insights into the long-term therapeutic benefits of replacing defective hematopoietic systems with healthy, young donor cells in a preclinical model of age-related disease.

3. The transplanted BM ages, but its correction remains effective. While the transplanted BM ages along with the recipient, the data highlight that correcting the intrinsic genetic defect in the hematopoietic system can have lasting therapeutic benefits. The persistence and functional capacity of the engrafted WT cells over time suggest that youthful, healthy BM can compete effectively with defective cells and mitigate disease progression, even in an aging environment. This finding aligns with the study's broader focus on strategies to improve hematopoietic function in aging-related contexts.

4. Relevance to the study's overarching focus. Although the primary focus of our study is on the engraftment and differentiation of young HSCs in aged environments, this experiment complements that narrative by providing evidence that young donor BM can confer sustained functional benefits in a model of age-associated disease. It offers a broader perspective on the therapeutic implications of our findings, particularly in preventing hematological disorders associated with aging.

We summarized the rationale for selecting 2 months as a treatment age in lines 287-291.

Reviewer #4 (Remarks to the Author):

We thank the reviewer for taking the time to review our manuscript.